Brief Communication

# Millennial-aged peat carbon outgassed by large humic lakes in the Congo Basin

Travis W. Drake [1] ✉, Jordon D. Hemingway [2], Matti Barthel [1], Antoine de Clippele [1], Negar Haghipour[2,3], Jose N. Wabakanghanzi[4], Kristof Van Oost [5] & Johan Six [1]

Congo Basin lakes Mai Ndombe and Tumba are major $CO_2$ sources. Here we show that their dissolved inorganic carbon is some 2,170–3,515 [14]C years old and partially (39–40%) originates from the surrounding peatlands. This implies a loss pathway for peat carbon, in which microbes respire old carbon within the peat and the resulting $CO_2$ is transported to the lakes and outgassed, linking these immense ancient stores to the modern carbon cycle.

Peatlands cover only ~3.8% of Earth's land surface[1], but contain approximately one-third (~600 PgC) of the world's total soil organic carbon (SOC)[2]. Recently, within the Congo Basin's central depression (that is, the 'Cuvette Centrale'), the largest known tropical peatland complex on Earth has been field verified and mapped[3], with an estimated C stock of 29 Pg (ref. 4). Congo peatlands are considered mostly pristine but they are still vulnerable to drainage and drying, whether through anthropogenic land-use change or shifts in climate. Indeed, recent paleoenvironmental analysis of peat cores from the Congo Basin show that past drying events have led to major losses of previously accumulated carbon stocks and that these peatlands today may lie close to a climatically driven drought threshold[5].

Within the Congo peat complex lie two massive humic lakes, Mai Ndombe (~2,250 km[2]) and Tumba (~700 km[2]) (Fig. 1a). Like other humic lakes, Mai Ndombe and Tumba receive inputs of both dissolved organic carbon (DOC), which fuels internal microbial respiration, and direct inputs of $CO_2$ from the surrounding highly productive swamp forests ecosystems. These inputs result in supersaturated concentrations of $CO_2$ relative to the atmosphere (1,700–3,100 ppm) and thus high $CO_2$ emissions (29–60 mmol m$^{-2}$ d$^{-1}$)[6,7].

It is generally understood that $CO_2$ emitted from humic lakes is derived from the decomposition of modern organic carbon (OC) from surrounding ecosystems and not from destabilized OC released from deep soil layers (that is, peat) where it has accumulated for millenia. So far, radiocarbon dating of $CO_2$ in ponds and streams within mid- and high-latitude peatland ecosystems has supported this view, showing that $CO_2$ is dominated by contemporary carbon[8–10]. The only exception

so far has been in streams draining disturbed peatlands, where aged OC has been observed[11–13].

To examine the source and age of $CO_2$ outgassing from the Tumba–Ngiri–Maindombe lake–peatland system, we analysed the isotopic composition of dissolved inorganic carbon (DIC) in Mai Ndombe and Tumba lakes. We discovered that DIC in both lakes was surprisingly old, with a mean radiocarbon age of $2170^{+208}_{-203}$ [14]C years (F[14]C = 0.76 ± 0.02) and $3515^{+387}_{-369}$ [14]C years (F[14]C = 0.65 ± 0.03) for Mai Ndombe and Tumba, respectively (Fig. 1b and Extended Data Table 1). By contrast, DOC for both lakes and particulate organic carbon (POC) for Mai Ndombe (POC was not measured in Tumba) were both modern (Fig. 1b and Extended Data Table 1). An aged DIC signature was also observed in the Fimi River, which receives the outflow from Lake Mai Ndombe. While the Fimi is a mixed system that also receives ~43% of its discharge from the Lukenie River, its similarly old DIC radiocarbon age still corroborates the lake DIC age and confirms the export of this ancient carbon from the lake system (Fig. 1b and Extended Data Table 1).

DIC in both lakes was [13]C depleted ($\delta^{13}$C = −23.5 ± 1.1‰ and −18.5 ± 1.2‰ for Mai Ndombe and Tumba, respectively) (Fig. 1b and Extended Data Table 1). The depleted lake DIC [13]C signature, combined with the absence of carbonate geology in the lake catchments, indicates an origin from terrestrial $C_3$ plant matter. While the bulk DIC pools are enriched by an average of 4.5–11.7‰ relative to their $C_3$-derived organic sources (probably due to $CO_2$ outgassing and/or atmospheric equilibration; Fig. 1b), a Miller-Tans plot[14] provides further support for this $C_3$ origin, exhibiting a strong linear relationship ($R^2 = 0.97$; Extended Data Fig. 1) and a slope of −28.7 ± 3.1‰, a value consistent

[1]Department of Environmental Systems Science, ETH Zurich, Zurich, Switzerland. [2]Geological Institute, Department of Earth and Planetary Sciences, ETH Zurich, Zurich, Switzerland. [3]Laboratory for Ion Beam Physics, Department of Physics, ETH Zurich, Zurich, Switzerland. [4]Department of Soil Physics and Hydrology, Congo Atomic Energy Commission, Kinshasa, Democratic Republic of Congo. [5]Earth and Life Institute, UCLouvain, Louvain-la-Neuve, Belgium. ✉e-mail: draketw@gmail.com

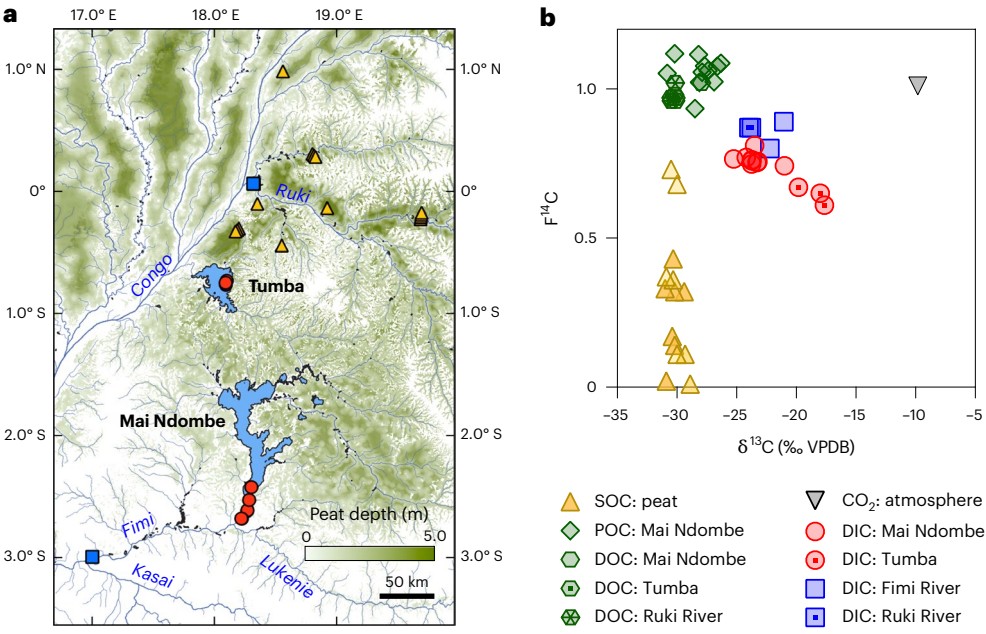

**Fig. 1 | Study sites and isotopic evidence for ancient peat carbon. a**, A map of the central Congo Basin showing sampling locations for lakes (Mai Ndombe and Tumba; red circles) and rivers (Fimi and Ruki; blue squares). The location of left-bank peat core sampling sites from ref. 16 are also shown (yellow triangles). The basemap indicates the peatland extent and depth (green gradient) based on data from ref. 4. **b**, A dual-isotope plot of radiocarbon activity ($F^{14}C$; values of 1 represent modern, <1 indicate progressively older carbon) versus stable carbon isotope ratios ($\delta^{13}C$). The colours of the data points correspond to the sampling locations shown in **a**, except for DOC and POC which have the same locations as DIC. Data in **b** are shown for Lake Mai Ndombe and Tumba DIC (red circles

and red circles with dots, respectively), Mai Ndombe POC (green diamonds) and Mai Ndombe and Tumba DOC (green hexagon and green hexagons with dots, respectively). River samples include Fimi River DIC (blue squares) and Ruki River DIC (blue square with dot) and DOC (hatched green hexagons). The potential sources shown in **b** include regional peat SOC (yellow triangles, darkening with increasing depth to 6 m)[16] and atmospheric $CO_2$ (inverted black triangle; Methods). The distinct isotopic position of the lake DIC (reds) clearly demonstrates a substantial contribution from ancient peat (yellows), contrasting with the modern DOC and POC pools (greens).

with $C_3$ OC as the primary DIC source. Even with this enrichment relative to DOC/POC and peat, this specific isotopic combination is, however, uncommon in the global context. While aged DIC is common globally, a recently compiled database shows it is typically associated with higher $\delta^{13}C$ values, a pattern consistent with inorganic carbon inputs from carbonate weathering[15]. The Congo lake DIC—being both aged and relatively $^{13}C$ depleted—is a clear exception to this global trend (Extended Data Fig. 2). These results strongly suggest that the DIC partially originates from the decomposition of ancient peat, which is known to have similar isotopic characteristics[16].

To quantify the contribution of peat to DIC, we performed a Monte Carlo simulation using both a two- and three-component isotopic mixing model (to simulate 100% kinetic fractionation due to outgassing and 100% atmospheric equilibration, respectively) that accounted for modern OC (that is, DOC and POC), ancient peat and atmospheric exchange. Our models converged and resulted in statistically consistent fractional peat contributions within uncertainty for each lake. The final weighted-average estimates indicate that ancient peat carbon accounts for 39 ± 8% of the DIC in Mai Ndombe and 40 ± 6% in Tumba (Extended Data Fig. 3). This confirms that the degradation of old, stored carbon is a major source of $CO_2$ evasion from lakes in this critical ecosystem. Combined with $CO_2$ outgassing flux estimates for Lake Mai Ndombe[7], this result implies that upwards of 150 Gg of peat C are outgassed each year by Lake Mai Ndombe alone. Importantly, this Mai Ndombe estimate is a conservative flux for peat C from the lake, as it does not include the lateral flux to the Fimi River nor the subsequent outgassing losses downstream, both of which are also substantially composed of peat-derived $CO_2$ (~27 ± 9%) (Extended Data Fig. 3).

These results give rise to the question of how this ancient, buried peat carbon ends up as $CO_2$ in the lake. Given the modern age of the POC

and DOC pools, it is unlikely that the respiration of this bulk OC within the lake is the source of the ancient DIC. Such a pathway would require a specific scenario: that the total OC input contains an ancient, exceptionally biolabile subpool that is preferentially and near-completely respired, leaving behind only the modern, less-labile pool that we measured. We find this scenario unlikely as both the ancient peat and the modern organic matter originate from the same $C_3$ forest vegetation, providing no clear biogeochemical reason why the older, more decomposed material would be more labile than the fresh material.

Instead, we propose a decoupling of the modern and ancient carbon pools, depicted in our conceptual model (Fig. 2). This model posits that modern, recently photosynthesized OC and its respiratory $CO_2$ are produced at the surface (Fig. 2, green pathways), while ancient peat OC is respired deeper within the surrounding peatland matrix and then transported via subsurface flows to the lake (Fig. 2, yellow arrows). Our model outlines three potential subsurface pathways: (1) aerobic respiration, (2) direct $CO_2$ production by acetoclastic methanogenesis and/or (3) hydrogenotrophic methanogenesis followed by $CH_4$ oxidation. While the dominant pathway remains unconstrained, the predominantly anoxic nature of the peatlands suggests that methanogenesis (pathways 2 or 3) is more likely than aerobic respiration (1), as the latter would require extensive drainage to create oxic conditions. We acknowledge, however, that the precise hydrological pathways and fluxes that transport this respired $CO_2$ are a key uncertainty and ongoing work to characterize the hydrology of the Congo peatlands will be crucial for resolving these complex dynamics. The proposed model framework is consistent with findings demonstrating a similar DOC and DIC age decoupling within boreal peatlands[17]. There, porewater DOC was shown to be modern while porewater DIC and $CH_4$ exhibited intermediate ages between the ancient peat and modern

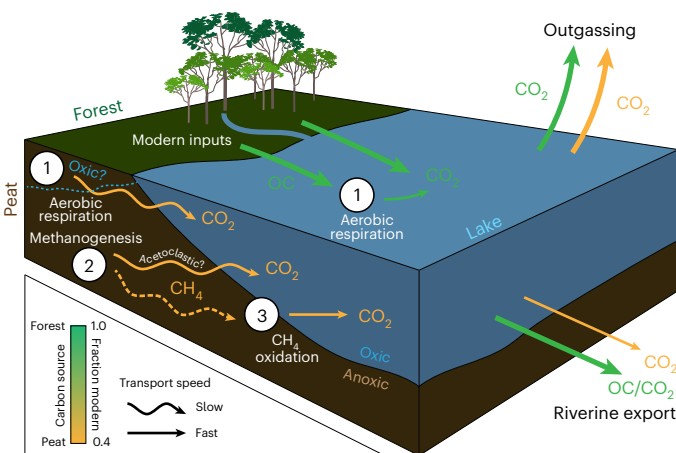

**Fig. 2 | Conceptual model of aged and modern carbon pathways to Lakes.** This schematic illustrates potential microbial processes generating $CO_2$ from two distinct sources. Pathways originating from the ancient peat carbon stock are shown in yellow ($F^{14}C$ -0.4), while pathways from the modern surface forest vegetation are in green ($F^{14}C$ -1.0). The arrows also indicate relative transport speed: straight arrows represent faster pathways (for example, surface flow) while wavy arrows represent slower pathways (for example, subsurface flow). The figure displays three potential pathways for aged $CO_2$ generation from peat: (1) aerobic respiration, (2) methanogenesis (producing aged $CO_2$ and methane ($CH_4$)) and (3) the oxidation of the aged $CH_4$ derived from methanogenesis. This aged $CO_2$ is transported to the lake, where it mixes with modern $CO_2$ produced from (1) the aerobic respiration of modern forest-derived OC. The lake then acts as a 'chimney', releasing this mixed pool of aged and modern $CO_2$ via outgassing and riverine export.

DOC. Similarly, the DOC from porewaters in tropical peatlands has been shown to be modern, even when the surrounding peat matrix is ancient[18,19]. These studies lend support to our decoupling model and further suggest that the ancient DIC in the lakes is sourced from deep subsurface respiration (probably involving methanogenesis) of ancient peat. Regardless of the specific pathway, once the aged $CO_2$ reaches the lake it is efficiently vented to the atmosphere, which makes the lake a large 'chimney' for respired old peat C.

These findings challenge the prevailing understanding that $CO_2$ emissions from pristine humic lakes are derived from modern, rapidly cycling carbon. Moreover, the slightly aged DIC in the Ruki River ($F^{14}C = 0.87$; Fig. 1b), a large blackwater tributary to the Congo[20], indicates the outgassing of aged $CO_2$ may be a broader feature of humic waters in the Cuvette Centrale. This discovery of a regional leakage of slow-cycling peat C represents a previously unrecognized mobilization pathway for this massive reservoir of OC to the atmosphere. It also implies that these stores of peat are not inert and that some fraction of the peat is being decomposed, even in the absence of direct anthropogenic disturbance. Whether this loss of peat C is a natural phenomenon—balanced by even greater inputs of stabilized OC in the peatlands—or indicative of more broadscale destabilization remains, however, unknown. Regardless, this finding amplifies concerns about the future stability of Congo peatlands. Given the current aquatic flux of aged $CO_2$, future peatland drying or drainage from climate or land-use change would probably accelerate this release by exposing deeper peat layers to oxygen and enhancing hydrologic connectivity. Indeed, if the peatlands already sit near a critical climate threshold[5], this pathway may become increasingly important.

Our discovery of a natural pathway for the release of millennial-aged peat C links a vast reservoir of ancient OC to the modern atmosphere. This 'lake-as-chimney' mechanism may arise owing to the unique hydrological setting of the Congo Basin, but if it is a more universal phenomenon, it may be occurring in other major peatland regions with large, integrated lakes (for example, Hudson Bay lowlands

and West Siberian lowlands). Either way, future work needs to reassess the stability of these critical systems and to incorporate this slow-cycle leak into global climate models.

## Online content

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

## Methods

### Site description

Lakes Mai Ndombe and Tumba are large (2,250 and 740 km[2], respectively), shallow (average depths ~4 m) lakes in the Democratic Republic of Congo, located within the Tumba–Ngiri–Maindombe wetland complex. Both lakes are surrounded by periodically and permanently flooded equatorial swamp forests underlain by thick peat deposits[3,4]. Mai Ndombe's watershed (48,900 km[2]) is rain fed, with major tributaries entering from the north and a single outflow, the Fimi River, at its southern point. Given an average annual outflow of ~1,295 m[3] s[−1], we estimate a water residence time of approximately 80 days. Tumba's watershed (7,100 km[2]) discharges directly into the Congo River and is also rain fed; however, there is probably exchange with the Congo during the wet season.

### Sampling protocol

Water samples in lake Mai Ndombe ($n = 13$) were collected at various depths using a custom-built, gas-tight Van Dorn sampler during expeditions in August 2022 (dry season) and January 2024 (wet season). For Lake Tumba ($n = 3$) and the Fimi ($n = 2$) and Ruki ($n = 3$) rivers, water samples were taken from 0.3 m below from the water surface. Subsampling for DIC, DOC and POC followed specific procedures.

For DIC measurements, 6 ml of water was collected into a headspace-free syringe. The water was immediately filtered through a 0.22-μm hydrophilic PTFE/L syringe-tip filter and injected into preprepared 12 ml exetainer vials. These vials were preflushed with $N_2$ and contained 50 μl of 50% $ZnCl_2$ (poison) and 50 μl of $H_3PO_4$ (acid). Triplicate vials were collected for concentration and another triplicate for $\delta^{13}C$ analysis.

For DOC, approximately 1 l of water was collected in high-density polyethylene (HDPE) bottles (prerinsed three times with sample water). The water was filtered through precombusted (450 °C for 4 h) and preweighed glass fibre filters (Whatman GF/F, 0.7 μm) using an acid-leached vacuum filter tower. The filtrate was then aliquoted into acid-leached opaque HDPE bottles and immediately acidified to pH 2 with 6 N HPLC-grade HCl for DOC analysis.

For POC, the filter from the DOC filtration step, which trapped the particulate matter from a known volume of water (~1 l), was retained. The filters were subsequently freeze dried, reweighed and stored for POC analysis.

### Analyses

The concentrations of different carbon pools were determined using distinct analytical techniques. DIC concentration was measured by quantifying the $CO_2$ gas in the headspace of pre-acidified sample vials using a gas chromatograph (Scion 456 GC) equipped with a thermal conductivity detector. DOC concentrations were determined via high-temperature catalytic oxidation on a Shimadzu TOC-L analyser, following the methods of ref. 21. POC concentrations were quantified by combusting aliquots of known area of the material collected on GF/F filters in an elemental analyser (varioMICRO cube, Elementar), as described in ref. 22.

For isotopic analyses ($\delta^{13}C$ and $^{14}C$), carbon from each pool was first converted to pure $CO_2$ gas. For DIC, the carbon was already present as $CO_2$ in the exetainer vials following acidification at the time of sampling. For POC, GF/F filter aliquots were first decarbonated using HCl vapour and then combusted to $CO_2$ in an Elementar Vario ISOTOPE select Elemental Analyzer (EA). For $^{14}C$-DOC, samples were concentrated via freeze-drying and for both $^{13}C$- and $^{14}C$-DOC, the DOC was converted to $CO_2$ via wet chemical oxidation[20].

The $CO_2$ gas derived from DIC and DOC was then analysed for its stable and radiocarbon isotopic compositions. For $\delta^{13}C$ analysis, the gas was analysed via isotope ratio mass spectrometry (IRMS), using a Finnigan Gasbench II coupled to a Finnigan Delta$^{plus}$ XP IRMS (with data corrected according to ref. 23) and an EA-coupled Isoprime precisION

IRMS for POC. All $\delta^{13}C$ results are reported in per mille (‰) relative to the Vienna Pee Dee Belemnite (VPDB) standard. For $^{14}C$ analysis, the $CO_2$ derived from DIC, DOC and POC was analysed using a mini carbon dating system accelerator mass spectrometer[24,25]. Radiocarbon results are reported as fraction modern (F$^{14}C$) following the conventions of ref. 26.

### End-member mixing analysis

To quantify the proportional contributions of different sources to the DIC pool, we used an end-member mixing analysis within a Monte Carlo framework. The analysis was performed in R using the tidyverse package. We defined three potential sources (end members) for DIC: (1) aged carbon from surrounding peat soils ('peat'), with an isotopic signature defined by SOC reported in ref. 16; (2) contemporary carbon from in-lake and riverine processing ('modern'), with an isotopic signature defined by the respective measured DOC and POC for each lake or river; and (3) atmospheric $CO_2$ ('atmosphere'), with a signature defined by established literature values corrected for fractionation due to dissolution ($\delta^{13}C = -9.80 \pm 0.10$‰; F$^{14}C = 1.01 \pm 0.01$)[14,27,28]. The measured DIC pool for a given water body was treated as the mixture.

We evaluated two distinct conceptual models to account for different gas-exchange dynamics. The first, an 'equilibration' model, assumed DIC was a three-way mixture of peat, modern and atmosphere sources, using both stable carbon isotopes ($\delta^{13}C$) and radiocarbon (F$^{14}C$) as tracers. The second, an 'outgassing' model, assumes strong $CO_2$ evasion prevents appreciable atmospheric equilibration[29], treating DIC as a two-part mixture of only peat and modern sources. As isotopic fractionation during outgassing affects $\delta^{13}C$, this second model relied exclusively on F$^{14}C$, which is, by definition, fractionation corrected[30]. For each model, 100,000 simulations were run. In each iteration, a value for each tracer was randomly drawn for the end members and the DIC mixture from normal distributions defined by their respective means and standard deviations. The system of linear equations was solved for the proportional contribution of each source, and only physically realistic solutions (that is, proportions between 0 and 1) were retained. This process yielded posterior probability distributions, from which we calculated the mean contribution and standard deviation for each source under both scenarios. For each lake, the mean peat contributions from both models were found to be statistically consistent within their uncertainty. We therefore calculated a final, single contribution for each lake using a weighted average, where each model's mean was weighted by the inverse of its posterior variance (1/s.d.[2]).

### Ethics and inclusion statement

This research was conducted in accordance with the Global Code of Conduct for Research in Resource-Poor Settings. We collaborated closely with the Régie des Voies Fluviales (RVF, DR Congo) to establish the study's local relevance and execute field campaigns, with additional partnership from the International Institute of Tropical Agriculture (IITA, Nigeria) and the Woodwell Climate Research Center (USA). All fieldwork operated under scientific research permits granted by the Ministère de l'Environnement et Développement Durable, DR Congo. This environmental research did not involve human participants, and researcher safety protocols were managed with guidance from our national RVF partners and the Kauka vessel crew. No biological materials or traditional knowledge were transferred out of the country.

## Data availability

All geochemical data generated in this study are provided in Extended Data Table 1. Peat thickness data used in Fig. 1 were obtained from Crezee et al.[4]. Atmospheric $CO_2$ isotope records used for end-member definition were obtained from Hua et al.[27] and the Scripps $CO_2$ Program.

## Code availability

The R source code used for the end-member mixing analysis and figure generation is available via Zenodo at https://doi.org/10.5281/zenodo.18017104 (ref. 31).

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

## Acknowledgements

This study was part of a project that received funding from the Swiss National Science Foundation under the Sinergia project 'Tropical Soil Erosion Dynamics (TropSEDs)' (CRSII5 205998). The funders had no role in study design, data collection and analysis, decision to publish or preparation of the manuscript. We thank the crew members and staff of the Régie des Voies Fluviales for providing their expertise in working along the Congolese waterways. We thank N. Mokwele Bey and G. Itoko for sampling the lake Tumba. Finally, we thank the Ministère de l'Environnement et Développement Durable, DR Congo and the local communities for granting us access to lakes Mai Ndombe and Tumba.

## Author contributions

T.W.D., J.D.H. and M.B. conceived of the study. T.W.D., J.D.H., M.B., A.d.C. and J.N.W. conducted the lake sampling. T.W.D., M.B. and N.H. generated the data. T.W.D., J.D.H. and M.B. analysed the data. T.W.D. wrote the paper with input from J.D.H., M.B., A.d.C., J.S. and K.V.O.

## Competing interests

The authors declare no competing interests.

## Additional information

**Extended data** is available for this paper at https://doi.org/10.1038/s41561-026-01924-3.

**Correspondence and requests for materials** should be addressed to Travis W. Drake.

**Extended Data Table 1 | Sample location, date, and carbon data (concentration, δ¹³C, and F¹⁴C) for Dissolved Inorganic Carbon, Dissolved Organic Carbon, and Particulate Organic Carbon**

| SITE | LAT | LONG | DEPTH | DATE | DIC | | | DOC | | | POC | | |
|---|---|---|---|---|---|---|---|---|---|---|---|---|---|
| | | | | | Conc. | δ¹³C | F¹⁴C | Conc. | δ¹³C | F¹⁴C | Conc. | δ¹³C | F¹⁴C |
| | (DD) | (DD) | (m) | (dd/mm/yyyy) | (mg L⁻¹) | (‰) | - | (mg L⁻¹) | (‰) | - | (mg L⁻¹) | (‰) | - |
| Mai Ndombe Lake | −2.618 | 18.269 | 0.05 | 13/8/2022 | 0.83 | −23.22 | 0.76 | 34.98 | −27.94 | 1.02005 | – | – | 0.93 |
| Mai Ndombe Lake | −2.618 | 18.269 | 4.00 | 13/8/2022 | 0.87 | −23.29 | 0.75 | – | – | – | – | – | |
| Mai Ndombe Lake | −2.613 | 18.271 | 0.05 | 22/2/2024 | 1.04 | −23.75 | 0.76 | 34.89 | – | – | 0.61 | −28.2 | 1.11 |
| Mai Ndombe Lake | −2.613 | 18.271 | 3.00 | 22/2/2024 | 1.10 | −23.75 | 0.75 | 35.10 | – | – | 0.65 | −26.3 | 1.09 |
| Mai Ndombe Lake | −2.613 | 18.271 | 6.00 | 22/2/2024 | 1.28 | −25.46 | – | 35.31 | – | – | 0.68 | −27.4 | 1.06 |
| Mai Ndombe Lake | −2.423 | 18.303 | 0.05 | 22/2/2024 | 0.94 | −24.19 | 0.77 | 35.88 | – | – | 0.62 | −30.2 | 1.12 |
| Mai Ndombe Lake | −2.423 | 18.303 | 6.00 | 22/2/2024 | 1.18 | −25.25 | 0.77 | 38.17 | – | – | 0.55 | −26.6 | 1.08 |
| Mai Ndombe Lake | −2.529 | 18.285 | 0.05 | 22/2/2024 | 1.00 | −21.57 | – | 34.89 | – | – | 0.51 | −27.7 | 1.08 |
| Mai Ndombe Lake | −2.529 | 18.285 | 3.00 | 22/2/2024 | 1.29 | −23.72 | 0.76 | 33.46 | – | – | 0.48 | −26.9 | 1.02 |
| Mai Ndombe Lake | −2.682 | 18.225 | 0.05 | 22/2/2024 | 1.15 | −20.98 | 0.74 | 33.30 | – | – | 0.44 | −27.9 | 1.06 |
| Mai Ndombe Lake | −2.681 | 18.221 | 0.05 | 24/2/2024 | 1.15 | −23.84 | – | 34.92 | −28.37 | – | 0.52 | −28.1 | 1.02 |
| Mai Ndombe Lake | −2.681 | 18.221 | 6.00 | 24/2/2024 | 1.28 | −23.95 | – | 34.86 | −28.43 | – | 0.41 | −30.8 | 1.05 |
| Mai Ndombe Lake | −2.681 | 18.221 | 10.00 | 24/2/2024 | 1.36 | −23.50 | 0.81 | 33.23 | −27.78 | – | – | −28.5 | 0.93 |
| Tumba Lake | −0.763 | 18.094 | 0.05 | 19/8/2025 | 0.72 | −19.82 | 0.67 | 22.32 | −30.12 | 0.96 | – | – | – |
| Tumba Lake | −0.732 | 18.101 | 0.05 | 19/8/2025 | 0.62 | −17.98 | 0.65 | 18.72 | −30.09 | 0.97 | – | – | – |
| Tumba Lake | −0.749 | 18.093 | 0.05 | 19/8/2025 | 0.59 | −17.65 | 0.61 | 20.08 | −30.20 | 0.98 | – | – | – |
| Fimi River | −2.995 | 17.001 | 0.05 | 10/8/2022 | 0.86 | −21.68 | 0.80 | 20.92 | −28.84 | 1.04 | – | −30.1 | 0.98 |
| Fimi River | −2.995 | 17.001 | 0.05 | 19/2/2024 | 4.07 | −20.48 | 0.89 | 17.05 | −28.84 | – | – | – | 0.92 |
| Ruki River | 0.062 | 18.322 | 0.05 | 20/8/2025 | 1.99 | −23.96 | 0.87 | 20.79 | −30.46 | 0.96 | – | – | – |
| Ruki River | 0.064 | 18.320 | 0.05 | 20/8/2025 | 2.06 | −24.29 | – | 19.10 | −30.48 | 0.97 | – | – | – |
| Ruki River | 0.065 | 18.318 | 0.05 | 20/8/2025 | 2.21 | −23.76 | 0.87 | 20.12 | −30.13 | 1.02 | – | – | – |

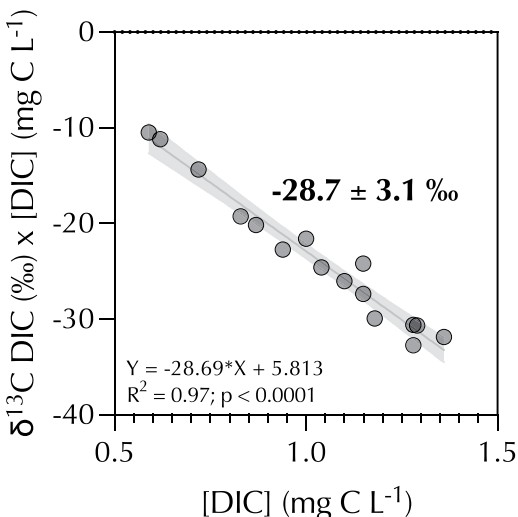

**Extended Data Fig. 1 | Miller-Tans plot of dissolved inorganic carbon (DIC) data from Lake Mai Ndombe and Lake Tumba.** The plot shows the product of $\delta^{13}$C-DIC and [DIC] versus [DIC]. Data from both lakes are combined, assuming shared isotopic sources and processes. The solid line represents a simple linear regression fitted to the data (n = 16; $R^2$ = 0.97, p = $1.32 \times 10^{-11}$), which has a slope of −28.7‰ and a y-intercept of +5.8. The shaded error band represents the 95% confidence interval of the regression. The slope value is interpreted as the DIC source value[14].

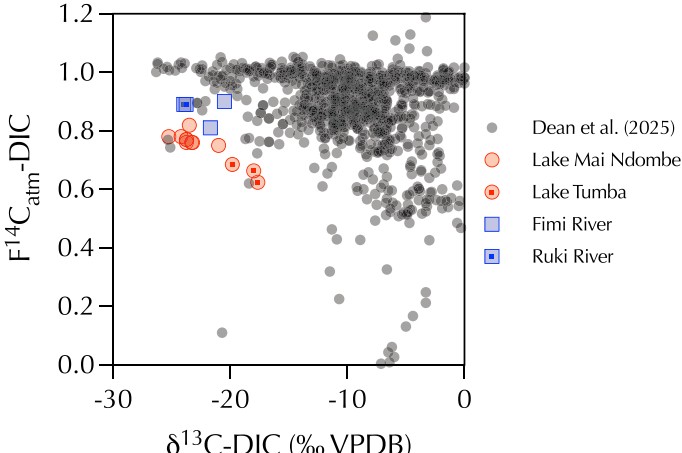

Legend:
- Dean et al. (2025)
- Lake Mai Ndombe
- Lake Tumba
- Fimi River
- Ruki River

**Extended Data Fig. 2 | Isotopic comparison of Congo Basin DIC with global riverine data.** Dual-isotope plot of dissolved inorganic carbon (DIC), showing atmosphere $^{14}CO_2$ normalized Fraction Modern ($F^{14}C_{atm}$) versus $\delta^{13}C$. Data from this study include Congo Basin lakes (red circles) and Congo Basin rivers (blue squares). Comparative data are from a global riverine dataset[15] (grey points). The global data were vetted; potential transcription errors noted in a few data points sourced from Ref. 32 within Ref. 15 compilation were excluded from the plot to ensure accuracy. The Congo Basin lake data form a distinct cluster characterized by highly depleted $\delta^{13}C$ and low $F^{14}C$ values, indicating a source from aged organic carbon. This signature is isotopically distinct from the global riverine DIC, which overwhelmingly falls within the mixing triangle defined by modern C3/C4 respiration, atmospheric $CO_2$ and carbonate weathering.

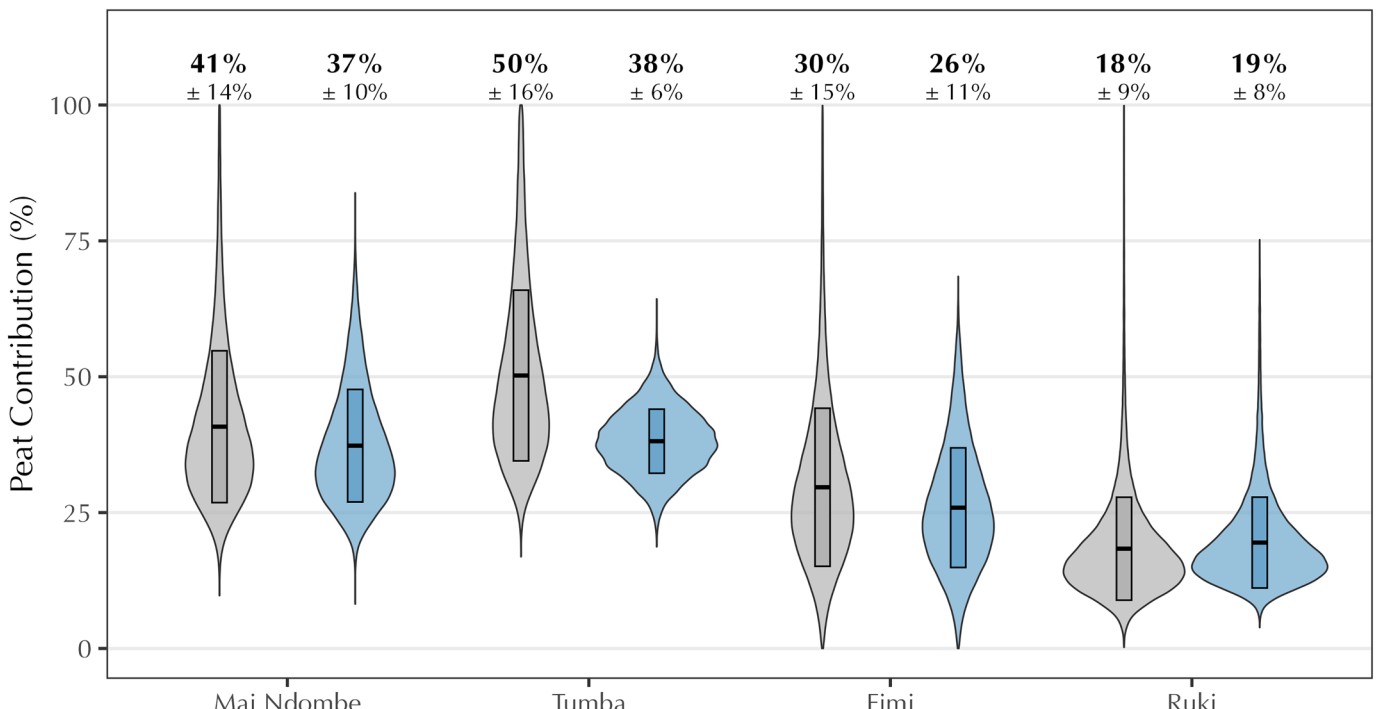

**Extended Data Fig. 3 | Monte Carlo analysis of ancient peat carbon contribution to DIC.** Probability distributions from the end-member mixing model simulations for each location. The analysis was run using two bounding scenarios: a two-source Outgassing model (grey violins) and a three-source Equilibration model (blue violins). Violin plots show the full probability distribution for the percent contribution from ancient peat derived from n = 100,000 independent Monte Carlo simulations. Input parameters for the measured DIC mixture distributions were derived from independent water samples collected from spatially distinct locations or depths for Lake Mai Ndombe (n = 9), Lake Tumba (n = 3), the Fimi River (n = 2), and the Ruki River (n = 2). Internal crossbars show the mean (center line) and ±1 standard deviation (ends of the bar). The mean (larger text) and ±1 standard deviation (smaller text) are also reported above each violin.

