## [Peer Review File · Nature Geoscience]

Millennial-aged peat carbon outgassed by large humic lakes in the Congo Basin

Corresponding Author: Dr Travis Drake

Version 0:

Decision Letter:

10th Oct 2025

Dear Dr Drake,

Your manuscript entitled "Africa's largest humic lake emits millennial-aged peat carbon" has now been seen by 3 referees, whose comments are attached.

The referees acknowledge the potential interest of your work, but between them, they also raise a number of concerns, which must prevent us from offering to publish the paper in its present form. It is not clear to us whether you will be able to address all the concerns raised. If you would like to pursue publication in Nature Geoscience, we will therefore need to see your responses to the criticisms raised and to some editorial concerns, along with a revised manuscript, before we can reach a decision regarding publication.

The referees' reports seem to be quite clear. Naturally, we will need you to address all of the points raised (that is, either to incorporate the suggestions or provide a compelling argument why the point made by the referee is not valid, or relevant to the editorial threshold as outlined below). Specifically, for publication in Nature Geoscience to be appropriate, we will need you to convincingly justify the interpretation of how ancient carbon, originally sequestered in the surrounding peatlands, ends up as CO₂ in the lake. You also need to offer plausible support for the broader implications of this process for the global carbon cycle under climate change.

In the light of the reports, it is not clear to us whether you will be able to meet this threshold, but if you feel that you will be able to do so, you will also need to make some formatting changes so that the revised manuscript is in Nature Geoscience style. These requirements are outlined below.

We hope to receive your revised paper within three weeks. If you cannot send it within this time, please let us know.

Please resubmit electronically using the link below to access your home page:

Link Redacted

*This url links to your confidential homepage and associated information about manuscripts you may have submitted or be reviewing for us. If you wish to forward this e-mail to co-authors, please delete this link to your homepage first.

Nature Geoscience is committed to improving transparency in authorship. As part of our efforts in this direction, we are now requesting that all authors identified as 'corresponding author' on published papers create and link their Open Researcher and Contributor Identifier (ORCID) with their account on the Manuscript Tracking System (MTS), prior to acceptance. This applies to primary research papers only. ORCID helps the scientific community achieve unambiguous attribution of all scholarly contributions. You can create and link your ORCID from the home page of the MTS by clicking on 'Modify my Springer Nature account'. For more information please visit www.springernature.com/orcid.

Yours sincerely,

FORMAT REQUIREMENTS:

TITLE: The title should give a sense of the main new findings, within 90 characters (including spaces). Please avoid using active verbs and punctuation.

LENGTH: The main text plus introductory paragraph (excluding Methods, references and figure legends) is usually no longer than 2200 words, with up to 4 display items (figures or tables). At the editor's discretion, we can allow up to 3000 words and 6 display items if required.

TEXT FORMAT: The text, including references, figure legends and acknowledgements, should be double-spaced and in 12 pt. font. Please include line numbers.

INTRODUCTORY PARAGRAPH (ABSTRACT): Please pay particular attention to the wording of the opening bold paragraph. This serves both as an introduction and as a brief, non-technical summary in about 180-200 words. Please start with two to three sentences on the motivation and scientific background to your work; use a phrase like "Here we present" to introduce your own work, including a brief description of methods and data; and follow on with an outline of your main findings, conclusions and their implications. This paragraph should not cite references and its contents should not be repeated elsewhere in the paper. Because scientists from other sub-disciplines will be interested in your results and their implications, it is important to explain essential but specialised terms concisely. We suggest that you show your introductory paragraph to colleagues in other fields to check for problematic concepts.

SUBHEADINGS: Articles should contain about 3 or 4 subheadings that break up the main text. Subheadings should be informative, should not contain punctuation, and must not exceed 58 characters (including spaces). There should not be a subheading immediately following the introductory paragraph.

METHODS: We offer an online-only Methods section of up to 3000 words (without figures or tables) that will be fully integrated with the HTML and PDF versions of the online paper, but not appear in the print version. This section can contain references that do not count towards the recommended 50 references for the main text, and will be fully indexed. References not in the main paper should be placed in a separate Methods references section after the Methods section, and numbered following on from the main reference list.

REFERENCES: As a guideline, Articles allow up to 50 references. Please check that your references conform to Nature Geoscience style and include either start and end page numbers or article numbers if page numbers are not available.

CORRESPONDING AUTHOR: Please include a statement after the references naming the author(s) to whom correspondence and requests for materials should be addressed, with an appropriate email address.

ACKNOWLEDGEMENTS: Please place the Acknowledgements section after the corresponding author details. Any submission detailing new material from protected sites should include information regarding the requisite permission obtained in this section. See <http://www.nature.com/nature-research/editorial-policies/reporting-standards> and <http://www.nature.com/articles/ngeo2587> for more information. Please note that reviewers are thanked in a separate Reviewer Recognition statement in our published papers.

AUTHOR CONTRIBUTIONS: We require authors to include a statement of their individual contributions to the paper – such as experimental work, project planning, data analysis, etc. – immediately after the acknowledgements. The statement should be short and refer to authors by their initials. See <https://www.nature.com/nature-research/editorial-policies/authorship> for more information.

ETHICS & INCLUSION STATEMENT: As part of efforts to promote greater equity in global research collaborations, we encourage researchers to follow the recommendations of the Global Code of Conduct for Research in Resource-Poor Settings in the design, execution and reporting of research and to provide a disclosure statement in the manuscript that considers these issues. Any "Ethics & Inclusion Statement" can be placed in the Methods section. Please see <https://www.nature.com/ngeo/editorial-policies/authorship#authorship-inclusion-and-ethics-in-global-research> for the full guidance on how to compose a statement.

FIGURES: For your reference, please see our figure guidelines (<https://www.nature.com/documents/NRJs-guide-to-preparing-final-artwork.pdf>).

FIGURE STYLE: Please avoid the use of uneven colour scales (e.g. rainbow-like) and colour schemes that are not accessible for colour vision deficient readers (e.g. red-green). See <https://rdcu.be/b9IIE> for a discussion of problems associated with some colour usage and suggestions for alternatives.

FIGURE CAPTIONS: All figure captions must begin with a short (up to 15–20 word) title sentence that describes the overall

content of the figure. Figure captions must provide a brief description of the figure and the symbols used and define any error bars in the figure, ideally within 100 words.

DATA SOURCES: We require all new data associated with the paper to be deposited in a persistent repository where they can be freely and enduringly accessed (see <https://www.nature.com/articles/s41561-019-0506-4>). We recommend submitting the data to discipline-specific, community-recognized repositories where possible (a list of recommended repositories is provided here: <http://www.nature.com/sdata/policies/repositories>). If a community resource is unavailable, data can be submitted to generalist repositories such as figshare (<https://figshare.com/>) or Dryad Digital Repository (<http://datadryad.org/>). Please provide a unique identifier for the data (for example a DOI or a permanent URL) in the Data availability statement, if possible. If the repository does not provide identifiers, we encourage authors to supply the search terms that will return the data. When selecting your repository, please note that repositories vary in how long they take to archive data.

Please refer to our data policies here: <http://www.nature.com/authors/policies/availability.html>

DATA AVAILABILITY STATEMENT: We require a paragraph as separate section after Methods but before Methods references entitled "Data availability", where any available information on data resources used in or produced for the paper is provided. For data that have been obtained from publically available sources, please provide a DOI or URL and the specific data product name. For data sets with a DOI, we additionally encourage full citation of the data in the reference list of the Methods section.

For more information on and an example from Nature Geoscience of how to compose the statement, please see: <http://www.nature.com/authors/policies/data/data-availability-statements-data-citations.pdf>

CODE AVAILABILITY STATEMENT: We strongly encourage authors to share computer code if possible. When code is central to the main conclusions of a manuscript, we additionally mandate that a statement is included under the heading "Code availability", indicating whether and how the code can be accessed, including any restrictions to access. Code availability statements should be provided as a separate section after the data availability statement but before the Methods references.

EXTENDED DATA: Up to 10 additional figures or tables can be published as Extended Data. Please ensure that these are cited in the main text (not only in the Methods or figure captions). Extended Data allows key figures and tables to be made more visible, as they will be integrated into the full-text HTML version of your paper and will be appended to the online PDF. Extended Data must be of similar quality to the main figures and tables and supplied as figures in JPG, TIF, or EPS format at a size that will allow both the figure/table and caption to be presented on a single A4 page.

SOURCE DATA: We encourage you to provide source data for your Figures and Extended Data (that is, data that are plotted in the figures) whenever possible. To facilitate reuse, we additionally encourage presentation of data in large tables in an editable format (such as Excel or CSV). For imaging source data, we encourage deposition in a relevant repository, such as figshare (<https://figshare.com/>) or the Image Data Resource (<https://idr.openmicroscopy.org>).

SUPPLEMENTARY INFORMATION: Please submit any additional Supplementary Information as a separate, single PDF document; this may contain text, tables or figures. Supplementary data (including any source data for the Supplementary Information) can be uploaded as separate files in Excel, ASCII or CSV format.

CONSORTIA AUTHORSHIP: For papers containing one or more consortia, all members of the consortium who contributed to the paper must be listed in the paper. If necessary, individual authors can be listed in both the main author list and as a member of a consortium listed at the end of the paper. When submitting your revised manuscript, the consortium name should be entered as an author in the manuscript tracking system, together with the contact details of a nominated consortium representative. See <https://www.nature.com/authors/policies/authorship.html> for our authorship policy and <https://www.nature.com/documents/nr-consortia-formatting.pdf> for further consortia formatting guidelines, which must be adhered to prior to acceptance.

WHEN REVISING YOUR PAPER, PLEASE

* Include a point-by-point response to any editorial suggestions and to our referees. Please include your response to the editorial suggestions in your cover letter, and please upload your response to the referees as a separate document.

* Include a tracked changes version of the manuscript, if possible

* Ensure it complies with our format requirements as set out in our guide to authors at <https://www.nature.com/ngeo/submission-guidelines>

* State in a cover note the length of the text, methods and legends; the number of references; number and estimated final size of figures and tables

* Ensure that all correspondence is marked with your Nature Geoscience reference number in the subject line.

REVIEWER COMMENTS:

Reviewer #1 (Remarks to the Author):

This paper presents the data from a C isotope survey at the watershed scale for the Lake Mai Bombe, within the Congo basin, Africa. The measurements include a peat depth profile (soil organic carbon), particulate and dissolved organic carbon in the lake water, together with dissolved inorganic carbon in the lake and river outlet. The study is important given the low number of field observations from African regions. The logistical challenges associated with carrying field work in these regions are immense. Most global studies identify the African continent as an area lacking observations, but with a potentially high relevance to the global C cycle and the role of inland waters within this cycle. The data presented here is unique and of good quality. The data analysis (MonteCarlo mixing model) is suitable. The interpretation of the data is accurate, although interpreting this kind of data can never be too precise. I think the core of the paper is suitable for publication in Nature Geoscience, but there can be more work done to improve the presentation of the data and make the paper richer, despite its short format.

In its current form, the paper appears rather basic - a data reporting from an underrepresented region. I believe more can be done to present this data in a more impactful way for the readers. As a reader, I would like to be able to better grasp:

1) just how important are the lake CO₂ emissions in this watershed. There is no concentration measurements presented in the figures. How large is the DIC pool compared with the DOC pool in this lake? Is the DIC pool mostly in CO₂ form, I would expect so with the humid content of the lake. Can the points in figure 2 be scaled to the concentrations? I would also suggest combining figure 1 and 2 in a two-panel figure to help the reader associate the sample with its location.

2) the interpretation of the processes leading to the observed differences in DOC and DIC cycling at the landscape level could be made clearer. The explanations for the age patterns are presented in the text (Line 84-86), but the sentence is hard to understand – I had to reread it several times. Would a schematic be useful here? Something like that of Duvert et al., (2019) Nature Geoscience (<https://doi.org/10.1038/s41561-018-0245-y>) would help the reader understand the possible scenarios to explain the data.

Essentially, the results highlight a disconnect between DOC and DIC cycling, with DIC being surprisingly old. This disconnect can arise from many processes - biological and hydrological. I agree that the possibility of an old-labile fraction of DOC causing these persistent old DIC values is unlikely. However, one key possibility is that an old-deep groundwater source, having old DIC but little DOC, was not captured in the sampling. This could supply old CO₂ and allow peat derived DOC to dominate the age signature of DOC in the lake. At this large spatial scale, I would not expect DIC generated in the peatland to make its way to the lake. It would be mostly evaded to the atmosphere along the way, before reaching the lake. Ultimately, the results show that the hydrological flux of DOC and DIC sources in this landscape are disassociated in time, despite both originating from C₃ plant substrate.

Along those lines, I find the term “peat” too precise when referring to the DIC source. It draws an association with the peatlands that were sampled in the study, while other OC reservoirs exist in the watershed. I believe the term “biogenic” or “terrestrial” would be more appropriate since it agrees with the δ¹³C-values of CO₂ (close to C₃ plant) without assuming that it comes from the geographic location of the peatland dominated parts of the watershed.

3) I would also like to see how the observed ¹⁴C values fit within a broader understanding of DOC and DIC cycling in other. The text makes rather few connections to the literature. Despite the short format of the paper, I believe this should be feasible and would strengthen the paper. Supplementary files provide figures to explain the monte-Carlo analysis. This supports well the methodology, but more figures could be placed in this section to put the findings in context. One would imagine putting the DOC and DIC radiocarbon data in a global context, for example from the data of <https://doi.org/10.1002/2014GB004911> and <https://doi.org/10.1038/s41586-025-09023-w>, which highlights the lack of data from the studied region. Just how unusual are these data in a global context? A density diagram presenting in background the global data, super imposing the data from this study would be ideal.

The author claims that there is a “prevailing understanding that CO₂ emissions from pristine humic lakes are derived from modern, rapidly cycling carbon”. I don’t agree. I would rather claim that there is a natural expectation that DOC and DIC should be cycled on similar timescales because they are mostly loaded together from terrestrial environments. Also, we would expect the lateral C flux (if supplied by biogenic sources) to be mostly contemporary simply because it makes more sense from a C budget perspective - all C fluxes in a given watershed operate on the same timescale, what was stored long ago remains stored etc. but its not.

Is the term “advected” (Line 84) perhaps too precise? It would imply only horizontal movement, while the possibility of a deeper/older groundwater source would imply both vertical and horizontal direction.

In line 97 the authors associate the destabilization of Congo peatlands with the presence of an old DIC pool in the lake. I don’t find those claims to be well supported by the data. There is no mechanistic evidence that this age patterns is caused by destabilization of the peatlands, and not just a normal property of this landscape.

Reviewer #2 (Remarks to the Author):

The manuscript by Drake and colleagues deals with the topic of the age of carbon dioxide transported into, and emitted from, Lake Mai Ndombe in the Congo Basin. They combine their own measurements with those from the literature to show that CO₂ within the lake is old, in contrast to POC and DOC which are younger. Measurements from the river draining the lake are also old. The authors suggested this aged CO₂ is derived from the respiration of old soil OC within the peat soils in the wider catchment. This finding is interesting, and unexpected, in that studies often show young CO₂ released from waterbodies, as recently fixed C is rapidly returned to the atmosphere. As the authors write, this finding might represent a destabilisation of Congo Basin peat stores, or it may simply be natural.

The paper is very well written, and engagingly so. Figures are clear. Discussion is supported by the results. I think the paper will be of interest to a general audience and find no major flaws with it.

One more substantial comment relates to the sampling strategy. I imagine the logistical difficulties of sampling these systems must be immense, so we should perhaps give some leeway. That said, it is a shame there aren't samples from the northern (peat-rich) end of the lake – what's the residence time within the lake / how old is the water at the outflow? Would you expect a change across the lake? Similarly, the peat SOC samples appear to be from a different catchment which is probably fine but a bit messy.

A few line comments follow:

L42. "The prevailing hypothesis in the literature is that CO₂ emitted from humic lakes is derived from decomposition of modern OC."

Admittedly it is a synthesis of all waterbody types, not just humic waters, but perhaps worth including mention of Dean et al (2025) and their new findings?

"Here we combine new and published measurements to create a global database of the radiocarbon content of river dissolved inorganic carbon (DIC), CO₂ and CH₄. Isotopic mass balance of our database suggests that $59 \pm 17\%$ of global river CO₂ emissions are derived from old carbon (millennial or older), the release of which is linked to river catchment lithology and biome."

"CO₂ derived from within-river heterotrophic and autotrophic respiration... can be millennial in age if derived from deeper soil respiration flushed to rivers laterally by hydrological flow paths."

<https://www.nature.com/articles/s41586-025-09023-w>

Figure 2. It's interesting that one of the Fimi River sites sits slightly away from the other (and the lake data points). Are the Fimi River sites replicates or from different locations?

L94. "Whether this loss of peat C is a natural phenomenon—balanced by even greater inputs of stabilized OC in the peatlands—or indicative of more broadscale destabilization remains, however, unknown."

In a way the authors should be applauded for their restraint here, but I am itching to know what their thoughts are. Do you think this is a destabilisation, driven by either changing climate or land-use change within the catchment (if there is any)? If you're too uncertain to say either way, then fine to leave as written already.

L110. What are the DOC, POC, and DIC concentrations in the lake? What's the residence time? What's the catchment size? Can you add these details please?

L129. Was there a reason 0.22 μ m was used for DOC rather than the commoner 0.45? Of course, you're also omitting a large fraction of OC here (0.22 – 0.7 μ m).

Reviewer #3 (Remarks to the Author):

Overall comments

Recent work has shown that Lake Mai Ndombe, in the DRC is a considerable source of CO₂ to the atmosphere. Here this study shows, through carbon isotopic composition that the CO₂ being outgassed is not modern, but on average ca. 2,200 years old, with this old carbon coming from the surrounding peatlands.

I think this is a well written paper presenting an interesting finding. I enjoyed reading it and thinking about the results. The methods used to show that this old dissolved inorganic carbon likely originates from the peatlands seem reasonable to me. However, I thought a weakness of the paper was the discussion around the different ages in POC/DOC and DIC (paragraph 7, lines 80-88). The finding of modern DOC/POC and ancient DIC was interesting and unexpected, and although I cannot immediately think of an obvious explanation for it, I wasn't totally convinced by the possible explanations put forward by the authors. I thought they could have laid out their conceptual models more clearly and discussed how their proposed mechanisms fit with what is already in the literature. I think it is ok to not have a confident answer to why the different carbon pools have different ages, but at the moment this section feels a bit rushed and incomplete. I give my thoughts on both conceptual models below in more detail, along with much more minor comments for the authors.

Overall, I think this is a nice paper, and the finding that DIC being outgassed from Mai Ndombe is old and likely coming from the peatlands is an interesting finding and makes an important contribution to the understanding of carbon cycling within the

Cuvette Centrale. Given the global importance of this region and the wide interest the Congo Basin peatlands tend to generate, I think this paper would be a good fit for a Nature Geoscience audience. If the authors gave a bit more consideration to the possible processes behind the different ages in the carbon pools, I'd be happy to see it published.

Conceptual models explaining age differences in DOC/POC and DIC

For the first possible explanation, I couldn't quite follow what was being said; I wondered if the authors could be more explicit by what they mean. I didn't understand what was meant by "direct inputs of peat OC"; do the authors mean DOC/POC? I can't think in what other form peat would enter the lake. I can't envisage chunks of peat entering the lake. So do the authors mean given that the average age of DOC/POC is modern, the ancient DIC is unlikely to be respired DOC/POC? And because I didn't understand this sentence, I couldn't follow what was being said about bulk peat and lability.

The second possible explanation, I understood to be that the old DIC is travelling to the lake via ground water, and modern POC, DOC and DIC is travelling by surface flow. Although we don't know the age of the peat around Mai Ndombe, peat carbon of ca. 2000 years old could come from within the top meter of the peat profile (see age depth models in Garcin et al. 2022, who are cited elsewhere in the paper) and I could envisage water tables dropping in dry season to this level now and again, but would this happen frequently enough to mean that enough old carbon is being transported to the lake, resulting in a mean DIC age of ca. 2000 years? Maybe. But if there was such considerable oxidation going on, I would then expect the DOC and POC to be equally old (please see Moore et al. 2013, who are cited later). Also, I wondered, would modern DOC and POC only travel via surface flow? A paper by Gandois et al. 2014 (<http://dx.doi.org/10.1016/j.gca.2014.03.012>) suggests not necessarily.

Could another possible explanation be methane oxidation? Methane is produced at depth which is then oxidised as it reaches the lake? I have no idea- just something that came to mind.

I also thought that a paper by Chanton et al. 2008 (doi:10.1029/2008GB003274) might be of interest to the authors and worth citing, as they also date DOC, DIC, and peat C, and find that DOC is modern, peat C is old, and the DIC age lies somewhere in between.

Minor comments

L13-15: Because I not sure the authors have exhausted all possible processes that could account for their results, I wonder if this should be stated in such a matter-of-fact way. Perhaps it would be good to present this more as the favoured working hypothesis.

L42: "The prevailing hypothesis in the literature"- would be good to have citations here.

L56-57: perhaps a brief explanation in the figure caption that F14C values of 1 represent modern, and less than 1 are older, to allow readers not familiar with F14C to interpret the figure.

L69: "To quantify this contribution": As it is a new paragraph it should be made clear that the authors are talking about ancient peat contribution to the isotopic signature.

Methods: I can't see any sample numbers anywhere. How many samples were taken per sample location? Were there replicates? Or are dates from single samples or composite samples? Just a bit more info about this would be nice.

Methods: It would be good if the authors added sample location coordinates somewhere.

*****END*****

Version 1:

Decision Letter:

3rd Dec 2025

Dear Dr Drake,

Your manuscript titled "Millennial-aged peat carbon emissions from Africa's largest humic lakes" (NGS-2025-09-03074A) has now been seen by our referees, whose comments are included below (note that one report is attached as a separate document). In light of the reviews, I am pleased to say that we can, in principle, offer to publish a revised version of your manuscript that addresses the referees' comments and complies with our editorial and formatting requirements.

We are now performing detailed checks on your paper. We should send you a checklist detailing our editorial and formatting requirements in about a week. Please do not finalise revisions or upload any final materials until you receive this additional

information from us. To get you started, I am including our generic formatting instructions at the end of this email. Feel free to contact me if you have any questions.

Yours sincerely,

Nature Geoscience

Research Cross-Journal Editorial Team

FORMAT REQUIREMENTS - BRIEF COMMUNICATIONS:

TITLE: The title should give a sense of the main new findings, within 90 characters (including spaces). Please avoid using active verbs and punctuation.

LENGTH: The main text plus introductory paragraph (excluding Methods, references and figure legends) should be no longer than 1500 words, with up to 2 display items (figures or tables).

INTRODUCTORY PARAGRAPH (ABSTRACT): Brief Communications include a brief, unreferenced abstract that provides a non-technical summary in up to 70 words (about 3 sentences). The abstract should concisely capture the motivation, methods, results and main conclusions of the work, ideally avoiding or explaining any specialist terminology. Acronyms should also be avoided.

SUBHEADINGS: Brief Communications do not have subheadings.

METHODS: We offer an online-only Methods section of up to 3000 words (without figures or tables) that will be fully integrated with the HTML and PDF versions of the online paper, but not appear in the print version. This section can contain references that do not count towards the recommended 20 references for the main text, and will be fully indexed. References not in the main paper should be placed in a separate Methods references section after the Methods section, and numbered following on from the main reference list.

REFERENCES: As a guideline, Brief Communications allow up to 20 references. Please check that your references conform to Nature Geoscience style (see <https://www.nature.com/ngeo/for-authors/preparing-your-submission#formatting>) and include either start and end page numbers or article numbers if page numbers are not available.

CORRESPONDING AUTHOR: Please include a statement after the references naming the author(s) to whom correspondence and requests for materials should be addressed, with an appropriate email address.

ACKNOWLEDGEMENTS: Please place the Acknowledgements section after the corresponding author details. Any submission detailing new material from protected sites should include information regarding the requisite permission obtained in this section. See <http://www.nature.com/nature-research/editorial-policies/reporting-standards> and <http://www.nature.com/articles/ngeo2587> for more information.

AUTHOR CONTRIBUTIONS: We require authors to include a statement of their individual contributions to the paper – such as experimental work, project planning, data analysis, etc. – immediately after the acknowledgements. The statement should be short and refer to authors by their initials. See <https://www.nature.com/nature-research/editorial-policies/authorship> for more information.

ETHICS & INCLUSION STATEMENT: As part of efforts to promote greater equity in global research collaborations, we encourage researchers to follow the recommendations of the Global Code of Conduct for Research in Resource-Poor Settings in the design, execution and reporting of research and to provide a disclosure statement in the manuscript that considers these issues. Any "Ethics & Inclusion Statement" can be placed in the Methods section. Please see <https://www.nature.com/ngeo/editorial-policies/authorship#authorship-inclusion-and-ethics-in-global-research> for the full guidance on how to compose a statement.

FIGURES: Choosing the right electronic format for your figures at this stage will speed up the processing of your paper. Please refer to our figure guidelines (<https://www.nature.com/documents/NRJs-guide-to-preparing-final-artwork.pdf>). If you are in doubt about the correct format for your figures after reading our guidelines, please ask the art editor for advice via geoscience@nature.com. We will edit your figures/tables so that they conform to Nature Geoscience style and will reproduce clearly in print. Please resize your figures to fit single or double column width (88 mm or 180 mm, respectively). If your figures contain several parts, the parts should be labelled lower case a, b, and so on, and form a neat rectangle when assembled. Please supply each figure in its entirety, NOT as separate panels.

FIGURE STYLE: Please avoid the use of uneven colour scales (e.g. rainbow-like) and colour schemes that are not accessible for colour vision deficient readers (e.g. red-green). See <https://rdcu.be/b91IE> for a discussion of problems associated with some colour usage and suggestions for alternatives.

FIGURE CAPTIONS: All figure captions must begin with a short (up to 15–20 word) title sentence that describes the overall content of the figure. Figure captions must provide a brief description of the figure and the symbols used and define any error bars in the figure, ideally within 100 words.

DATA SOURCES: We require all new data associated with the paper to be deposited in a persistent repository where they can be freely and enduringly accessed (see <https://www.nature.com/articles/s41561-019-0506-4>). We recommend submitting the data to discipline-specific, community-recognized repositories where possible (a list of recommended repositories is provided here: <http://www.nature.com/sdata/policies/repositories>). If a community resource is unavailable, data can be submitted to generalist repositories such as figshare (<https://figshare.com/>) or Dryad Digital Repository (<http://datadryad.org/>). Please provide a unique identifier for the data (for example a DOI or a permanent URL) in the Data availability statement, if possible. If the repository does not provide identifiers, we encourage authors to supply the search terms that will return the data. When selecting your repository, please note that repositories vary in how long they take to archive data.

Please refer to our data policies here: <http://www.nature.com/authors/policies/availability.html>

DATA AVAILABILITY STATEMENT: We require a paragraph as separate section after Methods but before Methods references entitled "Data availability", where any available information on data resources used in or produced for the paper is provided. For data that have been obtained from publicly available sources, please provide a DOI or URL and the specific data product name. For data sets with a DOI, we additionally encourage full citation of the data in the reference list of the Methods section.

For more information on and an example from Nature Geoscience of how to compose the statement, please see: <http://www.nature.com/authors/policies/data/data-availability-statements-data-citations.pdf>

CODE AVAILABILITY STATEMENT: We strongly encourage authors to share computer code if possible. When code is central to the main conclusions of a manuscript, we additionally mandate that a statement is included under the heading "Code availability", indicating whether and how the code can be accessed, including any restrictions to access. Code availability statements should be provided as a separate section after the data availability statement but before the Methods references.

TEXT FORMAT: The text, including references, figure legends and acknowledgements, should be double-spaced and in 12 pt. font.

EXTENDED DATA: Up to 10 additional figures or tables can be published as Extended Data. Please ensure that these are cited in the main text (not only in the Methods or figure captions). Extended Data allows key figures and tables to be made more visible, as they will be integrated into the full-text HTML version of your paper and will be appended to the online PDF. Extended Data must be of similar quality to the main figures and tables and supplied as figures in JPG, TIF, or EPS format at a size that will allow both the figure/table and caption to be presented on a single A4 page.

SOURCE DATA: We encourage you to provide source data for your Figures and Extended Data (that is, data that are plotted in the figures) whenever possible. To facilitate reuse, we additionally encourage presentation of data in large tables in an editable format (such as Excel or CSV). For imaging source data, we encourage deposition in a relevant repository, such as figshare (<https://figshare.com/>) or the Image Data Resource (<https://idr.openmicroscopy.org>).

SUPPLEMENTARY INFORMATION: Please submit any additional Supplementary Information as a separate, single PDF document; this may contain text, tables or figures. Supplementary data (including any source data for the Supplementary Information) can be uploaded as separate files in Excel, ASCII or CSV format.

CONSORTIA AUTHORSHIP: For papers containing one or more consortia, all members of the consortium who contributed to the paper must be listed in the paper. If necessary, individual authors can be listed in both the main author list and as a member of a consortium listed at the end of the paper. When submitting your revised manuscript, the consortium name should be entered as an author in the manuscript tracking system, together with the contact details of a nominated consortium representative. See <https://www.nature.com/authors/policies/authorship.html> for our authorship policy and <https://www.nature.com/documents/nr-consortia-formatting.pdf> for further consortia formatting guidelines, which must be adhered to prior to acceptance.

Referee comments:

Reviewer #1 (Remarks to the Author):

See attached PDF.

Reviewer #2 (Remarks to the Author):

Drake and colleagues have done a thorough job of responding to my comments. Particularly I appreciate them adding their DOC, DIC and POC concentrations; for the inclusion of the comparison with Dean et al; and for including new data from a second lake. For me, the manuscript is now acceptable for publication, and I look forward to seeing the published version.

Mike Peacock

Reviewer #3 (Remarks to the Author):

The authors have done a good job of addressing my concerns about the lack of a clear explanation around the decoupled DIC and DOC/POC ages. This part of the manuscript has been significantly improved- I now find it much easier to follow the proposed mechanisms, and figure 2 greatly helps the reader visualise what may be going on. I appreciate the additional citations and the inclusion of Extended Data Figure 2, as this gives much better context to the results. The addition of Extended Data Table 1 means the methods are now much more transparent. I liked the addition of the data from Lac Tumba and the Ruki River, and agree with the authors that this makes their manuscript much stronger. Overall, I find the manuscript greatly improved and would be happy to see it published.

I spotted some very minor proofreading issues:

L60: Typo- reads "Maindombe" instead of "Mai Ndombe"

L61: DIC should be spelled out here instead of L62.

Figure 2: Is the yellow CO₂ riverine export arrow meant to be smaller than the green arrow? It sort of suggests relative importance of pathways, which I don't think it is meant to.

Dear Reviewers,

We thank you for your detailed and constructive comments on our manuscript, now titled “Millennial-aged peat carbon emissions from Africa's largest humic lakes” [NGS-2025-09-03074]. Your comments and suggestions greatly improved the manuscript.

Before addressing the specific comments point-by-point below (reviewer comments in black; our responses in blue), we wish to first announce a significant addition to the revised manuscript: **we have now included new DIC and DOC data from Lake Tumba**, the other massive humic lake in the Cuvette Centrale. We wish to emphasize how this new dataset provides substantially stronger support for our central interpretation. It independently corroborates the Lake Mai Ndombe findings (notably, featuring even older DIC, but similarly modern DOC) and adds confidence in the robustness of our isotopic framework. Combined with a new conceptual figure and global comparison analysis, these additions result in a revised manuscript where we present a mechanistic synthesis that explains how and why ancient carbon is emitted from these tropical lakes. We believe these revisions directly elevate the manuscript from a regional data report into an impactful, mechanistic synthesis that provides readers with a clear framework to grasp this globally relevant process.

Sincerely,

Travis Drake
travis.drake@usys.ethz.ch
(on behalf of all authors)

Reviewer #1:

This paper presents the data from a C isotope survey at the watershed scale for the Lake Mai Bombe, within the Congo basin, Africa. The measurements include a peat depth profile (soil organic carbon), particulate and dissolved organic carbon in the lake water, together with dissolved inorganic carbon in the lake and river outlet. The study is important given the low number of field observations from African regions. The logistical challenges associated with carrying field work in these regions are immense. Most global studies identify the African continent as an area lacking observations, but with a potentially high relevance to the global C cycle and the role of inland waters within this cycle. The data presented here is unique and of good quality. The data analysis (MonteCarlo mixing model) is suitable. The interpretation of the data is accurate, although interpreting this kind of data can never be too precise. I think the core of the paper is suitable for publication in Nature Geoscience, but there can be more work done to improve the presentation of the data and make the paper richer, despite its short format.

We thank the reviewer for their positive assessment of the study's importance, data quality, analyses, and suitability for Nature Geoscience. We agree that the manuscript's impact can be enhanced by improving the data presentation and contextualization. We have addressed their individual suggestions below.

In its current form, the paper appears rather basic - a data reporting from an underrepresented region. I believe more can be done to present this data in a more impactful way for the readers. As a reader, I would like to be able to better grasp:

1) just how important are the lake CO₂ emissions in this watershed. There is no concentration measurements presented in the figures. How large is the DIC pool compared with the DOC pool in this lake? Is the DIC pool mostly in CO₂ form, I would expect so with the humid content of the lake. Can the points in figure 2 be scaled to the concentrations? I would also suggest combining figure 1 and 2 in a two-panel figure to help the reader associate the sample with its location.

The reviewer raises an important point about the overall importance of the lake's CO₂ emissions within the watershed's carbon balance. A complete landscape-scale C balance would be required to fully answer this question, but such an analysis, which involves summing diffuse CO₂ emissions from both terrestrial and aquatic systems, is beyond the scope of this study. However, aquatic systems are widely recognized as biogeochemical hotspots that disproportionately contribute to landscape-level emissions. Given Lake Mai-Ndombe's (and now also Lake Tumba's) large surface area, its known supersaturation with CO₂ (detailed in the cited Barthel et al. 2025), and its direct exposure to the atmosphere, the lakes are undoubtedly a critical component of the regional carbon budget. Our finding that a large portion of this flux originates

from ancient peat highlights a previously unquantified pathway for old carbon loss, making these specific lake emissions highly significant.

The reviewer's questions about the carbon pool concentrations are insightful. You are correct that the DIC pool is small compared to the DOC pool; the concentration data, which we have now included as **Extended Data Table 1**, shows a ratio of approximately 1:30. Your guess about the form of DIC is also correct. The lakes are humic and acidic, and as detailed in Barthel et al. (2025), over 90% of the DIC is present as dissolved CO₂.

We appreciate the suggestion to scale the points in the new Figure 1 by concentration, but we've opted not to do this as we feel it would be misleading. Our paper's central finding concerns the *fluxes* of carbon, not the *stocks*. While the DOC *stock* is ~30-fold larger, the *fluxes* are much closer in magnitude. This is because the DIC pool is an open system with rapid turnover from outgassing, whereas the DOC pool is primarily lost via riverine export. A calculation using outgassing (from Barthel et al. 2025) and riverine export data shows the total DOC flux is only ~1.95 times the total DIC flux. Scaling the figure by concentration (stock) would visually imply that DOC cycling (1:30) dominates DIC cycling, when the flux data (1:1.95) shows their contributions are comparable. We believe this would obscure the significance of the DIC flux, which is the "chimney" mechanism at the core of our paper. We prefer to present the raw isotopic data, which clearly shows the distinct sources, and provide the concentration data in **Extended Data Table** for full transparency.

Finally, we agree with your recommendation for unifying the figures. We have combined Figure 1 and Figure 2 into a new two-panel figure. As you noted, this helps the reader associate the isotopic data with the sampling locations and regional context. Thank you very much for this suggestion!

2) the interpretation of the processes leading to the observed differences in DOC and DIC cycling at the landscape level could be made clearer. The explanations for the age patterns are presented in the text (Line 84-86), but the sentence is hard to understand – I had to reread it several times. Would a schematic be useful here? Something like that of Duvert et al., (2019) Nature Geoscience (<https://doi.org/10.1038/s41561-018-0245-y>) would help the reader understand the possible scenarios to explain the data.

Essentially, the results highlight a disconnect between DOC and DIC cycling, with DIC being surprisingly old. This disconnect can arise from many processes - biological and hydrological. I agree that the possibility of an old-labile fraction of DOC causing these persistent old DIC values is unlikely. However, one key possibility is that an old-deep groundwater source, having old DIC but little DOC, was not captured in the sampling. This could supply old CO₂ and allow peat derived DOC to dominate the age signature of DOC in the lake. At this large spatial scale, I would not expect DIC generated in the peatland to make its way to the lake. It would be mostly

evaded to the atmosphere along the way, before reaching the lake. Ultimately, the results show that the hydrological flux of DOC and DIC sources in this landscape are disassociated in time, despite both originating from C3 plant substrate.

We thank the reviewer for this constructive feedback and for identifying the lack of clarity in our explanation. We have now rewritten the passage for clarity and, as suggested, have added a new conceptual figure (now **Figure 2**) to explicitly illustrate the hydrologic and temporal disconnect between DOC and DIC sources.

The new figure displays our central hypothesis: modern carbon (OC and CO₂) is sourced from the near surface, while the ancient CO₂ is produced and delivered via subsurface flow. We agree that the exact biogeochemical pathways generating this aged CO₂ are uncertain, and our figure therefore depicts several possibilities (aerobic respiration, methanogenesis, and methane oxidation) that could occur within the deep peat matrix.

The reviewer raises a critical point about the potential for CO₂ evasion during transport from the peatland to the lake. In this system, we conceive of "subsurface flow" as rainwater moving slowly and laterally as porewater *through the peat matrix itself*, which is a water-saturated, largely anoxic environment. Because this transport occurs below the surface within the peat, exchange with the atmosphere is likely limited, allowing dissolved gases to accumulate and be transported towards the lake. The lake then functions as a large "chimney"—a window to the atmosphere where this accumulated subsurface gas can be efficiently outgassed. This mechanism explains how CO₂ generated at a distance within the peatland can reach the lake without significant prior evasion.

We fully agree that the precise hydrological pathways and fluxes are a key uncertainty in this landscape. We have added a sentence to the manuscript acknowledging this (lines 133-136), and we note that ongoing work by our colleagues to characterize the hydrology of the Congo peatlands will be crucial for resolving these complex dynamics in the future.

Along those lines, I find the term "peat" too precise when referring to the DIC source. It draws an association with the peatlands that were sampled in the study, while other OC reservoirs exist in the watershed. I believe the term "biogenic" or "terrestrial" would be more appropriate since it agrees with the δ¹³C-values of CO₂ (close to C3 plant) without assuming that it comes from the geographic location of the peatland dominated parts of the watershed.

We appreciate the reviewer's point regarding the precision of our terminology. However, we have opted to retain the term "peat" as our isotopic data, combined with the landscape's known carbon stocks, provide strong, specific evidence that we feel precludes a more general "terrestrial" source.

The lake DIC is *millennial-aged* and the only C3 carbon reservoir in this watershed with a comparable age and the necessary massive scale is the surrounding peatland complex, which stores 29 Pg of ancient carbon. The only other potential source of aged C3 carbon would be old soil organic carbon (SOC) from the upland "terra firma" soils. However, mobilizing this aged SOC into the lake would require significant and widespread soil erosion. Given the region's low topographic relief and minimal anthropogenic disturbance, there is no evidence for erosion at the scale needed to explain our findings. Furthermore, the peatlands are, by definition, intimately and hydrologically connected to the lake system. Our proposed mechanism—whereby CO₂ is generated within the peat matrix and transported via subsurface porewater flow—provides a direct and plausible pathway for this specific aged carbon to reach the lake.

Therefore, while the source is certainly "biogenic" and "terrestrial," we feel that our data allow for a more precise identification. Using a general term would obscure the central finding of our paper: that the immense ancient peat stock is the most parsimonious source for the aged CO₂ being emitted from the lake.

3) I would also like to see how the observed ¹⁴C values fit within a broader understanding of DOC and DIC cycling in other. The text makes rather few connections to the literature. Despite the short format of the paper, I believe this should be feasible and would strengthen the paper. Supplementary files provide figures to explain the monte-Carlo analysis. This supports well the methodology, but more figures could be placed in this section to put the findings in context. One would imagine putting the DOC and DIC radiocarbon data in a global context, for example from the data of <https://doi.org/10.1002/2014GB004911> and <https://doi.org/10.1038/s41586-025-09023-w>, which highlights the lack of data from the studied region. Just how unusual are these data in a global context? A density diagram presenting in background the global data, super imposing the data from this study would be ideal.

We thank the reviewer for this constructive suggestion to place our findings within a broader global context. While the Short Communication format constrained our initial reference list, we agree that this comparison significantly strengthens the paper.

Following the reviewer's recommendation, we have added a new figure (**Extended Data Fig. 2**). This dual-isotope plot compares the ¹⁴C and ¹³C values of our lake DIC with the global riverine DIC dataset from Dean et al. (2025). We focused this new analysis on DIC, as it exhibits the most striking and unique isotopic signature in our study. In contrast, we find the DOC results—and the divergence between modern DOC and aged DIC—less globally unique, as this pattern is common in river systems where modern DOC from vegetation coexists with old DIC from carbonate weathering.

During this process, we also vetted the global data; for instance, we noted potential transcription errors in a few data points sourced from Chen et al. (2021) within the Dean et al. (2025) compilation and excluded them from our plot to ensure accuracy.

The resulting figure (**Extended Data Fig. 2**) clearly illustrates that the DIC from our study lakes occupies a distinct isotopic space. The data cluster shows a combination of old ^{14}C values with highly depleted ^{13}C signatures. This comparison highlights just how unusual this system is, as our data points to a source from preserved, old organic carbon. This signature is largely unrepresented in the current global dataset, which primarily reflects DIC from some mix of modern C3/C4 plant respiration, atmosphere, and carbonate rock weathering. We have added this figure to the Supplement and incorporated a discussion of this important context into the main text (Line 79-86).

Extended Data Fig. 2 | Isotopic comparison of Congo Basin DIC with global riverine data. Dual-isotope plot of dissolved inorganic carbon (DIC), showing atmosphere $^{14}\text{CO}_2$ normalized Fraction Modern ($F^{14}\text{C}_{\text{atm}}$) versus $\delta^{13}\text{C}$. Data from this study include Congo Basin lakes (red circles) and Congo Basin rivers (blue squares). Comparative data are from a global riverine dataset² (grey points). The global data were vetted; potential transcription errors noted in a few data points sourced from Ref.⁴ within Ref.² compilation were excluded from the plot to ensure

accuracy. The Congo Basin lake data form a distinct cluster characterized by highly depleted $\delta^{13}\text{C}$ and low $F^{14}\text{C}$ values, indicating a source from aged organic carbon. This signature is isotopically distinct from the global riverine DIC, which overwhelmingly falls within the mixing triangle defined by modern C3/C4 respiration, atmospheric CO_2 and carbonate weathering.

The author claims that there is a “prevailing understanding that CO_2 emissions from pristine humic lakes are derived from modern, rapidly cycling carbon”. I don’t agree. I would rather claim that there is a natural expectation that DOC and DIC should be cycled on similar timescales because they are mostly loaded together from terrestrial environments. Also, we would expect the lateral C flux (if supplied by biogenic sources) to be mostly contemporary simply because it makes more sense from a C budget perspective - all C fluxes in a given watershed operate on the same timescale, what was stored long ago remains stored etc. but its not.

We thank the reviewer for this thoughtful point, as it gets to the core of our study's novelty. Our original statement about the "prevailing understanding" was based on our survey of literature from other well-studied systems (e.g., boreal humic lakes), where CO_2 emissions are often dominated by the respiration of recent, rapidly-cycling terrestrial carbon. We agree with the reviewer's main premise: there is a natural expectation that DOC and respiratory CO_2 should be cycled on similar timescales, especially when they are loaded together from terrestrial environments. This expectation, and the common exceptions to it (such as the presence of old DIC from carbonate weathering, which we note is not a factor here), are precisely what frame the surprising nature of our results.

The reviewer's logic is correct—from a standard C budget perspective, one would expect fluxes to operate on similar timescales. Our findings are striking because they contradict this expectation. We observe a significant decoupling of these pools ages, driven not by geogenic inputs derived from the weathering of carbonate rocks, but by the unique mobilization of old, preserved peatland carbon that is processed and released as CO_2 . We have now clarified this point in the text.

Is the term “advected” (Line 84) perhaps too precise? It would imply only horizontal movement, while the possibility of a deeper/older groundwater source would imply both vertical and horizontal direction.

We thank the reviewer for their careful consideration of our terminology. The reviewer is correct that 'advected' strongly implies horizontal transport, and we appreciate their point that a deeper groundwater source would indeed involve vertical movement. We have now revised this section of the text to better describe our conceptual model, removing the use of “advected” and replacing it with the more neutral “transported”.

In line 97 the authors associate the destabilization of Congo peatlands with the presence of an old DIC pool in the lake. I don't find those claims to be well supported by the data. There is no mechanistic evidence that this age patterns is caused by destabilization of the peatlands, and not just a normal property of this landscape.

We thank the reviewer for this important point and we agree with their assessment. The reviewer is correct that our data identify the *presence* of an old DIC pool, but they do not, by themselves, provide direct mechanistic evidence that this is caused by a 'destabilization' of the peatlands, rather than being a 'normal' property of this landscape.

Our intention was to offer 'destabilization' as a possible, large-scale explanation, but we concede that the term was too conclusive and not fully supported by the current data.

To address this, we have revised this section (formerly Line 97) to remove this strong claim. We now clarify that 'destabilization' is one potential *driver* that could *enhance* the pathways shown in our new conceptual figure. For example, processes like peat draining (a form of destabilization) could introduce oxygen to deeper, older peat layers, thereby accelerating the 'aerobic respiration' pathway (Process 1 in the figure) and leading to the aged DIC we observed.

This revised framing better distinguishes between the processes we illustrate (in the conceptual figure) and the potential large-scale drivers (like destabilization vs. 'normal' steady-state function) that are not yet constrained by our data. We thank the reviewer for helping us clarify this important distinction.

Reviewer #2:

The manuscript by Drake and colleagues deals with the topic of the age of carbon dioxide transported into, and emitted from, Lake Mai Ndombe in the Congo Basin. They combine their own measurements with those from the literature to show that CO₂ within the lake is old, in contrast to POC and DOC which are younger. Measurements from the river draining the lake are also old. The authors suggested this aged CO₂ is derived from the respiration of old soil OC within the peat soils in the wider catchment. This finding is interesting, and unexpected, in that studies often show young CO₂ released from waterbodies, as recently fixed C is rapidly returned to the atmosphere. As the authors write, this finding might represent a destabilisation of Congo Basin peat stores, or it may simply be natural.

The paper is very well written, and engagingly so. Figures are clear. Discussion is supported by the results. I think the paper will be of interest to a general audience and find no major flaws with it.

We are grateful to the reviewer for their positive and encouraging assessment of our manuscript. We appreciate their feedback and hope they find that our revisions and point-by-point responses below have further strengthened the paper.

One more substantial comment relates to the sampling strategy. I imagine the logistical difficulties of sampling these systems must be immense, so we should perhaps give some leeway. That said, it is a shame there aren't samples from the northern (peat-rich) end of the lake – what's the residence time within the lake / how old is the water at the outflow? Would you expect a change across the lake? Similarly, the peat SOC samples appear to be from a different catchment which is probably fine but a bit messy.

We thank the reviewer for this comment, and we share their appreciation for the immense logistical challenges of sampling in this region. The reviewer is correct that logistical constraints prevented us from sampling the northern end of Lake Mai Ndombe. However, to substantially strengthen our study and broaden its regional context, we have now included data from an entirely new lake, Lake Tumba (see revised Figure 1). Lake Tumba is the second-largest humic lake in the region, and our new data show it possesses an *even older* DIC signature than Lake Mai Ndombe.

This major addition also helps to address the reviewer's valid point about the peat SOC samples. With the inclusion of Lake Tumba, some of the peat core locations from Dargie et al. (2025) now lie directly within the Lake Tumba catchment, strengthening the link between the aged peat stores and the aged DIC we observed in the water.

Regarding the reviewer's question about expected changes across Lake Mai Ndombe, we did not observe any consistent longitudinal patterns in the transect we sampled and new dissolved CO₂ concentration data from the northern shore of the lake are consistent with the regions we sampled (Barthel et al. 2025). This lack of a strong spatial gradient is consistent with the lake's hydrology. We performed a rough calculation of the water turnover time based on the lake's known volume and outflow:

Volume: Average Depth (~4 m) multiplied by Area (~2250 km²) = ~9 km³

Outflow (average of wet and dry season ADCP measurements): ~1295 m³ s⁻¹

Residence Time: 9,000,000,000 m³) divided by 1295 m³ s⁻¹ = ~ 6,950,000 seconds (**80 days**)

This rapid turnover suggests the lake is relatively well-mixed and that water at the outflow is (in terms of water age) quite young. This rapid flushing would likely prevent the formation of strong, persistent spatial gradients in dissolved carbon age across the lake.

A few line comments follow:

L42. “The prevailing hypothesis in the literature is that CO₂ emitted from humic lakes is derived from decomposition of modern OC.”

Admittedly it is a synthesis of all waterbody types, not just humic waters, but perhaps worth including mention of Dean et al (2025) and their new findings?

“Here we combine new and published measurements to create a global database of the radiocarbon content of river dissolved inorganic carbon (DIC), CO₂ and CH₄. Isotopic mass balance of our database suggests that $59 \pm 17\%$ of global river CO₂ emissions are derived from old carbon (millennial or older), the release of which is linked to river catchment lithology and biome.”

“CO₂ derived from within-river heterotrophic and autotrophic respiration... can be millennial in age if derived from deeper soil respiration flushed to rivers laterally by hydrological flow paths.”
<https://www.nature.com/articles/s41586-025-09023-w>

We thank the reviewer for this highly relevant suggestion. We completely agree that the Dean et al. (2025) paper provides the essential global context for our study. This point was also raised by another reviewer, and we have made this comparison a key part of our revision. We have now added the Dean et al. (2025) reference and incorporated it into the main discussion. Furthermore, we created a new figure (**Extended Data Fig. 2**) that directly plots our lake DIC (¹⁴C and ¹³C) against their global riverine dataset.

The new figure clearly illustrates that the vast majority of the global data from the Dean et al. (2025) compilation occupies an isotopic space consistent with known sources, such as modern respiration mixing with DIC from carbonate weathering. In contrast, our Congo lake data occupy a distinct isotopic space—being both old (low F¹⁴C) and highly ¹³C-depleted—which is largely unrepresented in the global dataset. As our new figure (**Extended Data Fig. 2**) shows, this unique signature falls far outside the mixing bounds for carbonate contributions, pointing unequivocally to the aged, preserved organic carbon (peat) source we hypothesize.

Figure 2. It’s interesting that one of the Fimi River sites sits slightly away from the other (and the lake data points). Are the Fimi River sites replicates or from different locations?

We thank the reviewer for this keen observation. The two Fimi River data points represent samples from the same location, but they were collected during two different seasons (wet and dry). The reviewer is correct to note that these points are distinct from the lake data. This is because the Fimi River is not simply the outflow of Lake Mai Ndombe; it is a mixed system that also receives inflow from the Lukenye River (which contributes ~43% of the Fimi's total discharge). The Lukenye River likely carries a different, and presumably younger, DIC signature. Therefore, the Fimi's isotopic value reflects a variable mix of the aged DIC from Lake Mai Ndombe and this other source. The difference between our two Fimi samples is thus explained

by seasonal changes in the contributions of these two river systems. We realize this is an important point of clarification and have now added this hydrological context to the text (line 68-71).

L94. “Whether this loss of peat C is a natural phenomenon—balanced by even greater inputs of stabilized OC in the peatlands—or indicative of more broadscale destabilization remains, however, unknown.”

In a way the authors should be applauded for their restraint here, but I am itching to know what their thoughts are. Do you think this is a destabilisation, driven by either changing climate or land-use change within the catchment (if there is any)? If you’re too uncertain to say either way, then fine to leave as written already.

We thank the reviewer for this comment and for appreciating our restraint on this point. It is a question we have discussed at length. We are hesitant to overstate our findings, as this leakage of old CO₂ could indeed be a natural, steady-state phenomenon. That said, the fact that this isotopic signature is so uncommon in the global context (**Extended Data Fig. 2**), even among studies that explicitly looked for it in northern peatlands, is precisely what gives us pause. This rarity raises the critical, unanswered question: are the Congo peatlands fundamentally different (e.g., uniquely large or dense enough to sustain this 'leak' as a balanced, steady-state loss) or is our observation indicative of a net loss? Given that our current data cannot distinguish between these two important scenarios, we feel the most scientifically sound approach is to maintain our original, cautious wording in the manuscript.

L110. What are the DOC, POC, and DIC concentrations in the lake? What’s the residence time? What’s the catchment size? Can you add these details please?

We thank the reviewer for requesting these additional details. We have now added **Extended Data Table 1**, which contains the concentration data. We have also now included the catchment size and approximate residence time (calculations shown above) in the lake in the methods (lines 172-178).

L129. Was there a reason 0.22µm was used for DOC rather than the commoner 0.45? Of course, you’re also omitting a large fraction of OC here (0.22 – 0.7 µm).

We thank the reviewer for this careful observation, which has allowed us to correct an error in the Methods section. The 0.22 µm filter pore size for DOC was mistakenly reported in the manuscript text, originating from a different protocol used for sampling large quantities of river sediments within our larger TropSEDS project. For the Lac Mai Ndombe samples presented in this paper, we instead collected smaller volumes and filtered them through 0.7 µm GF/F filters. This correction clarifies that our DOC is operationally defined as the 0.7 µm fraction, which is a common standard in aquatic biogeochemistry and resolves the reviewer's valid concern about the

0.22 – 0.7 μm fraction. We have now corrected this error in the Methods section.

Reviewer #3:

Overall comments

Recent work has shown that Lake Mai Ndombe, in the DRC is a considerable source of CO₂ to the atmosphere. Here this study shows, through carbon isotopic composition that the CO₂ being outgassed is not modern, but on average ca. 2,200 years old, with this old carbon coming from the surrounding peatlands.

I think this is a well written paper presenting an interesting finding. I enjoyed reading it and thinking about the results. The methods used to show that this old dissolved inorganic carbon likely originates from the peatlands seem reasonable to me. However, I thought a weakness of the paper was the discussion around the different ages in POC/DOC and DIC (paragraph 7, lines 80-88). The finding of modern DOC/POC and ancient DIC was interesting and unexpected, and although I cannot immediately think of an obvious explanation for it, I wasn't totally convinced by the possible explanations put forward by the authors. I thought they could have laid out their conceptual models more clearly and discussed how their proposed mechanisms fit with what is already in the literature. I think it is ok to not have a confident answer to why the different carbon pools have different ages, but at the moment this section feels a bit rushed and incomplete. I give my thoughts on both conceptual models below in more detail, along with much more minor comments for the authors.

We thank the reviewer for their positive feedback regarding the paper. We agree that the discussion of the disconnect between the OC and IC pools was a weakness in the original manuscript. In response to this comment and those from the other reviewers, we have thoroughly revised this section (formerly paragraph 7, lines 80-88). We have expanded our discussion, clarified our explanations, and included a new conceptual model (**Figure 2**) that we believe clearly illustrates our understanding. We hope these changes provide the clarity and depth the reviewer was looking for. We will address the reviewer's specific, detailed thoughts on our original conceptual models in the point-by-point response below.

Overall, I think this is a nice paper, and the finding that DIC being outgassed from Mai Ndombe is old and likely coming from the peatlands is an interesting finding and makes an important contribution to the understanding of carbon cycling within the Cuvette Centrale. Given the global importance of this region and the wide interest the Congo Basin peatlands tend to generate, I think this paper would be a good fit for a Nature Geoscience audience. If the authors gave a bit more consideration to the possible processes behind the different ages in the carbon pools, I'd be happy to see it published.

We thank the reviewer for their positive assessment, their supportive comments, and their endorsement of the paper's fit for Nature Geoscience. We are greatly encouraged by their feedback.

Conceptual models explaining age differences in DOC/POC and DIC

For the first possible explanation, I couldn't quite follow what was being said; I wondered if the authors could be more explicit by what they mean. I didn't understand what was meant by "direct inputs of peat OC"; do the authors mean DOC/POC? I can't think in what other form peat would enter the lake. I can't envisage chunks of peat entering the lake. So do the authors mean given that the average age of DOC/POC is modern, the ancient DIC is unlikely to be respired DOC/POC? And because I didn't understand this sentence, I couldn't follow what was being said about bulk peat and lability.

We apologize for the confusion this sentence caused. The reviewer's interpretation is exactly correct: by 'direct inputs of peat OC,' we were indeed referring to DOC and/or POC. As the reviewer guessed, our primary argument was that it is unlikely that the bulk DOC/POC pool, which we measured as modern, is the direct source of the lake's ancient DIC via respiration.

The subsequent discussion on 'bulk peat and lability,' which the reviewer correctly noted was confusing, was our attempt to explore the only hypothetical scenario under which this could be possible. We have now clarified this point in the text (lines 115-122), explaining that this scenario would require the total OC pool entering the lake to be a mix of ancient and modern carbon, where the ancient sub-pool is preferentially respired (i.e., because it is more biolabile) than the modern pool. This selective respiration would simultaneously produce the ancient DIC we observe while leaving behind the less-labile, modern DOC/POC pool (which is what we measured). While such age-lability relationships are known in other systems (e.g., permafrost), we find this explanation unlikely for our site. This is because both the ancient peat and the modern OC originate from the same C3 forest material, and there is no clear biogeochemical reason why the older peat material would be more labile than the fresh, modern OC. We have completely rewritten this paragraph to remove the ambiguous phrasing and explicitly lay out this 'preferential respiration' hypothesis and our counter-argument. We thank the reviewer for highlighting this critical point of confusion.

The second possible explanation, I understood to be that the old DIC is travelling to the lake via ground water, and modern POC, DOC and DIC is travelling by surface flow. Although we don't know the age of the peat around Mai Ndombe, peat carbon of ca. 2000 years old could come from within the top meter of the peat profile (see age depth models in Garcin et al. 2022, who are cited elsewhere in the paper) and I could envisage water tables dropping in dry season to this level now and again, but would this happen frequently enough to mean that enough old carbon is being transported to the lake, resulting in a mean DIC age of ca. 2000 years? Maybe. But if there was such considerable oxidation going on, I would then expect the DOC and POC to be equally

old (please see Moore et al. 2013, who are cited later). Also, I wondered, would modern DOC and POC only travel via surface flow? A paper by Gandois et al. 2014 (<http://dx.doi.org/10.1016/j.gca.2014.03.012>) suggests not necessarily.

Yes, the reviewer has correctly understood our explanation. Like the reviewer, we are also unsure of whether the water table fluctuations they describe (which is analogous to Pathway 1 in our new conceptual model) would be sufficient to result in our observed patterns. We agree with their key point: such a process would likely be accompanied by aged DOC/POC inputs as well, which is inconsistent with our data. We nevertheless include Pathway 1 as a possible, though perhaps non-dominant, mechanism by which aged peat-derived CO₂ could enter the lake.

We also thank the reviewer for the Gandois et al. (2014) reference. Gandois et al. indeed find modern DOC at depth in tropical peat, just as Chanton et al. (2008) did in boreal systems. This strongly supports our hypothesis that the DIC and DOC pools are decoupled. Our results, in concert with these studies, suggest that aged DIC is produced within the peat (even in the presence of modern DOC, as they show), and that both this aged DIC and the modern DOC are then mobilized to the lake, resulting in our observed age discrepancy.

Could another possible explanation be methane oxidation? Methane is produced at depth which is then oxidised as it reaches the lake? I have no idea- just something that came to mind.

Yes, that's an excellent point. We agree that methanogenesis at depth followed by subsequent methane oxidation in the lake is a highly probable explanation for our results. Indeed, Barthel et al. 2025 find evidence of strong CH₄ oxidation via the ¹³C isotopic signature of methane in Lake Mai Ndombe.

This pathway (where ancient CH₄ produced at depth is oxidized to aged CO₂) is now depicted as **Pathway 3** in our conceptual figure and discussed in the text (lines 128-133). This process would be perfectly consistent with our findings, as it provides a direct anoxic mechanism for transforming ancient solid peat into the aged, mobile DIC we observe in the lake.

I also thought that a paper by Chanton et al. 2008 (doi:10.1029/2008GB003274) might be of interest to the authors and worth citing, as they also date DOC, DIC, and peat C, and find that DOC is modern, peat C is old, and the DIC age lies somewhere in between.

We thank the reviewer for this highly relevant reference. The Chanton et al. (2008) paper is indeed a key study. As the reviewer notes, their findings in a boreal system mirror our own: they also document a striking decoupling where DOC is modern, but the DIC and CH₄ are intermediately-aged.

We have now cited this paper in the text (line 139), as it provides strong precedent for our conceptual model. It supports the exact mechanism we propose—that microbes (likely methanogens) within the peat are at least partially bypassing the modern DOC and respiring the ancient, solid-phase peat, thus producing the aged DIC that is eventually mobilized to the lake.

Minor comments

L13-15: Because I not sure the authors have exhausted all possible processes that could account for their results, I wonder if this should be stated in such a matter-of-fact way. Perhaps it would be good to present this more as the favoured working hypothesis.

We agree and have now changed the word “uncovers” to “implies” to tone down the certainty.

L42: “The prevailing hypothesis in the literature”- would be good to have citations here.

We agree with the reviewer. As written, that sentence implied citations were immediately needed.

Since the evidence supporting this general statement was in the *following* sentences (our citations 8-10), we have rephrased this topic sentence to "It is generally understood...". We then revised the next sentence to read "...has supported this view, showing..." to explicitly link this conventional wisdom to the evidence.

L56-57: perhaps a brief explanation in the figure caption that F14C values of 1 represent modern, and less than 1 are older, to allow readers not familiar with F14C to interpret the figure.

That's an excellent point, we thank the reviewer for the suggestion. We have now added a brief explanation of the F¹⁴C values directly into the now **Figure 1** caption as requested.

L69: “To quantify this contribution”: As it is a new paragraph it should be made clear that the authors are talking about ancient peat contribution to the isotopic signature.

We agree and have now changed the sentence to start with “To quantify the contribution of peat to DIC,…”

Methods: I can't see any sample numbers anywhere. How many samples were taken per sample location? Were there replicates? Or are dates from single samples or composite samples? Just a bit more info about this would be nice.

We apologize for the lack of information on sample numbers. We have now included the information in the methods text and in **Extended Data Table 1**.

Methods: It would be good if the authors added sample location coordinates somewhere.

We agree and have now included the GPS coordinates in the methods and **Extended Data Table 1**.

We thank the reviewers for their constructive feedback. We have listed the specific comments from this latest round of reviews below, followed by our responses in blue.

Reviewer #1 (Remarks to the Author):

I agree with the author's response. The new figure 2 is well crafted and will form an essential part of the paper. If I may suggest some visual improvements to this schematic, I would recommend that the author evaluate the possibility of separating the content of the image into at least two panels for better readability.

In its current form, there are many arrows pointing in many directions and its hard to extract information from the image. The trees in the back serve as a good entry point, but is this where the author want the reader to start? Despite the numbering, I did not know how to circulate through the image instinctively. Seperating the content somehow could make it easier to circulate through the content and identify the differences in mechanisms/cycling across timescales the author is presenting in the paper.

A simple solution would to replicate the images side by side, keeping the background catchment design constant, but modifying the arrows so the eye can spot the differences between the two images (timescales).

A more ambitious design, although it might require a certain level of graphical design skills, would be to create a sort of flat time animation. Everyone understands the concept of layers in animation, the author could use to visual concept to communicate overlapping timescales in C cycling. Could the schematic be replicated and overlapped in a sequence like this one: <https://docs.toonboom.com/help/harmony-20/advanced/gettingstarted/transform.html>

The paper reveals that DOC and CO₂ are cycled through different timescales in this landscape; DOC being more contemporary, while at least a portion of the CO₂/DIC is very old or "ancient". One could conceptualize this cycle to occur through many overlapping timescales (layers), with two extremes (young and old) dominating the cycle of one element in particular

Making the middle frames more transparent, and the two end frames opaque, would allow to visualize the two dominant timescales, and still integrate the idea that there is likely a mix of timescales in between contributing to aquatic C cycling, but they are less obvious than the extremes (contemporary and ancient).

I let the author determine the best course of action, but I think it's worth investing effort in this key figure.

We thank the reviewer for their thoughtful and creative comments on how to improve Figure 2. We agree in principle that separating the figure into multiple panels could isolate individual processes; however, we are concerned that doing so would require the figure to expand significantly to retain legibility.

We purposefully designed the figure in three dimensions to simultaneously depict surface and subsurface processes, highlighting how they occur in tandem but at different depths. We feel that separating these components into side-by-side panels would render the 3D depiction less impactful and obscure the connectivity of the system. Therefore, we believe the current

figure, in concert with the detailed caption and main text, provides the most efficient and clear depiction of the conceptual model.

While we appreciate the suggestion, we have decided to keep Figure 2 in its current form to preserve this holistic view—a decision supported by our internal review and the positive feedback from the other reviewers. However, we agree that a future study definitively identifying specific mechanisms would certainly benefit from the multi-layered visualization envisioned by the reviewer.

Reviewer #2 (Remarks to the Author):

Drake and colleagues have done a thorough job of responding to my comments. Particularly I appreciate them adding their DOC, DIC and POC concentrations; for the inclusion of the comparison with Dean et al; and for including new data from a second lake. For me, the manuscript is now acceptable for publication, and I look forward to seeing the published version.

Mike Peacock

Thank you for your reviews!

Reviewer #3 (Remarks to the Author):

The authors have done a good job of addressing my concerns about the lack of a clear explanation around the decoupled DIC and DOC/POC ages. This part of the manuscript has been significantly improved- I now find it much easier to follow the proposed mechanisms, and figure 2 greatly helps the reader visualise what may be going on. I appreciate the additional citations and the inclusion of Extended Data Figure 2, as this gives much better context to the results. The addition of Extended Data Table 1 means the methods are now much more transparent. I liked the addition of the data from Lac Tumba and the Ruki River, and agree with the authors that this makes their manuscript much stronger. Overall, I find the manuscript greatly improved and would be happy to see it published.

I spotted some very minor proofreading issues:

L60: Typo- reads “Maindombe” instead of “Mai Ndombe”

Fixed.

L61: DIC should be spelled out here instead of L62.

Fixed.

Figure 2: Is the yellow CO₂ riverine export arrow meant to be smaller than the green arrow? It sort of suggests relative importance of pathways, which I don't think it is meant to.

We appreciate the reviewer's attention to detail. The variation in arrow size is indeed intentional and serves as a semi-quantitative representation of the relative flux magnitudes. While we chose not to explicitly detail this scaling in the legend to avoid clutter, the specific proportional contributions are discussed in the main text. We prefer to retain the arrow

scaling to provide the reader with an immediate visual indication of the dominant pathways.